# Analysis and prediction of carbon storage changes on the Qinghai-Tibet Plateau

**Lei Wang[1], Yaping Zhang[1]\*, Xu Chen[2,3]\***

**1** School of Information Science and Technology, Yunnan Normal University, Kunming, China, **2** Faculty of Geography, Yunnan Normal University, Kunming, China, **3** The Engineering Research Center of Geographic Information System Technology in Western China, Ministry of Education, Yunnan Normal University, Kunming, China

\* zhangyp@ynnu.edu.cn (YZ); chenxu@ynnu.edu.cn (XC)

**Data availability statement:** All relevant data are within the manuscript and its Supporting information files.

**Funding:** This research was funded by two sources. The first source of funding was the Yunnan Provincial Agricultural Basic Research Joint Special Project, supported by the Yunnan Provincial Science and Technology Department, with Grant No. 202101BD070001-042. The second source of funding came from the Yunnan Ten-thousand Talents Program, supported by the Yunnan Provincial Department of Human Resources and Social Security.

## Abstract

The Qinghai-Tibet Plateau, a crucial global carbon reservoir, plays an essential role in the carbon cycle. This study used the Integrated Valuation of Ecosystem Services and Trade-offs (InVEST) model to analyze land use and carbon storage changes from 2000 to 2020, and the Patch-generating Land Use Simulation (PLUS) model to predict land use trends and carbon storage for 2030 and 2040 under various scenarios, combining carbon density data. The impact of driving factors on carbon storage and spatial heterogeneity were assessed using the Ordinary Least Squares (OLS) and Geographically Weighted Regression (GWR) models. Results showed a fluctuating increase in carbon storage, mainly from grasslands and forests, with soil organic carbon as the largest pool. Positive factors included Digital Elevation Model (DEM), temperature, proximity to railways, roads, and Normalized Difference Vegetation Index (NDVI), while aridity was negative. Predictions suggest carbon storage will rise across all scenarios, with ecological protection showing the largest increase. This study comprehensively analyzes the impact of climate and land use changes on carbon storage in the Qinghai-Tibet Plateau, enhances understanding of the plateau's ecosystem sustainability, and supports policy-making.

## 1 Introduction

The dangers of global warming have been increasingly evident since the 19th century [1], with more frequent extreme weather events like heatwaves, droughts, storms, and floods [2]. The accelerated melting of polar ice caps and glaciers has led to a rise in global sea levels, posing a great threat to coastal cities [3]. Global warming has had varying negative impacts on biodiversity, human health, and economic development [4–6]. Previous research indicates that the primary cause of global warming is the increase in greenhouse gas concentrations in the atmosphere, particularly carbon dioxide, methane, and nitrous oxides [7]. On the one hand, the burning of fossil fuels and industrial processes continuously release large amounts of greenhouse gases, exacerbating the greenhouse effect; on the other hand, activities such as deforestation and overgrazing have weakened the Earth's carbon sink capacity [8]. Climate changes further affect vegetation growth and microbial activity, indirectly impacting carbon sequestration and release [9]. Recent studies have shown that climate change and land-use

**Competing interests:** The authors have declared that no competing interests exist.

change are the main factors influencing changes in carbon storage [10–12]. Therefore, studying the impacts of climate and land-use changes on carbon storage is significant for reducing greenhouse gas emissions, enhancing Earth's carbon sink capacity, and ensuring sustainable development.

The Qinghai-Tibet Plateau is the world's highest and largest plateau, with an average elevation exceeding 4,000 meters, often called the "Third Pole of the World" [13]. The region experiences low temperatures year-round, leading to slow biological degradation processes, and its soil stores a large amount of carbon [14]. Due to its unique topography, the climate exhibits extreme characteristics with significant diurnal temperature variations, making the ecosystem highly sensitive to external changes [15]. Since the 1960s, the average temperature on the Qinghai-Tibet Plateau has increased by 0.2°C per decade, approximately twice the observed rate of global warming, and precipitation has also shown an increasing trend. According to the Fifth Assessment Report of the Intergovernmental Panel on Climate Change (IPCC), global warming is expected to continue shortly [16], which will further affect the dynamics of ecosystem carbon storage and exacerbate the uncertainty of carbon storage in the Qinghai-Tibet Plateau. Notably, the Qinghai-Tibet Plateau is a significant carbon reservoir in China, with more than 90% of its carbon stored in the soil [17]. Therefore, under the current context of global warming, conducting an in-depth analysis of the spatiotemporal patterns of carbon storage on the Qinghai-Tibet Plateau, assessing the specific impacts of climate change on carbon storage, and predicting its future dynamics not only help scholars gain a more comprehensive understanding of the sustainability of the Qinghai-Tibet Plateau ecosystem but also provide a scientific basis for policy-making. This is of great significance for protecting the ecological environment of the Qinghai-Tibet Plateau and addressing global climate change.

In recent years, as the quantitative assessment of carbon storage has gradually become a core focus in global climate change research, the related research methods have also developed rapidly. Currently, the main strategies for assessing carbon storage include ground surveys, remote sensing measurements, and model simulations [18–20]. Each of these methods has its advantages and limitations. For example, ground surveys provide high accuracy but require substantial cost and labor input; remote sensing technology offers broad regional coverage but lacks the precision of ground-level details [21]; while model simulations can maintain cost-effectiveness while providing wider spatial and temporal coverage, integrating multi-source data, and supporting complex decision analysis [22,23]. For instance, Chen et al. used the InVEST model to analyze the dynamics and influencing factors of the carbon reservoir in the Loess Plateau of China over the past 40 years [24]. Li et al. utilized the InVEST model along with different land use data and land cover scenarios to predict and analyze changes in carbon storage in regions such as Hangzhou and Chongqing [25–28]. Liang et al. combined the PLUS model with multi-objective planning to predict the land use structure under different optimized scenarios in Wuhan [29]. However, most of these studies have focused on densely populated or economically active areas, such as major cities in East Asia, Europe, and North America. In contrast, there has been little research on the Qinghai-Tibet Plateau, a region characterized by extreme natural conditions and relatively low human activity. Additionally, many studies examine the impacts of driving factors like climate change or land use change on carbon storage separately, without analyzing them. This creates limitations in comprehensively understanding how these factors interact in complex environments to influence the spatiotemporal changes in carbon storage.

Overall, to address the aforementioned limitations, the main tasks and contributions of this paper are as follows:

- Analyzed the spatiotemporal changes in land use and carbon storage on the Qinghai-Tibet Plateau from 2000 to 2020.
- Used the PLUS model to predict the land use distribution on the Qinghai-Tibet Plateau for 2030 and 2040, and applied the InVEST model to obtain the corresponding carbon storage distribution for these years.
- Identified significant driving factors influencing changes in carbon storage using the Ordinary Least Squares (OLS) method, and employed the Geographically Weighted Regression (GWR) model to explicitly represent the spatial non-stationarity and heterogeneity of these factors, exploring their impact on carbon storage changes.

This study aims to partially fill the gaps in existing carbon storage and land use prediction analysis methods and regional research, providing a reference for formulating land management strategies to adapt to climate change, protect ecological security, and promote sustainable development on the Qinghai-Tibet Plateau.

## 2 Datasets and research methods

### 2.1 Study area

The Qinghai-Tibet Plateau is located in southwestern China (25°59'37"N ~ 39°49'33"N, 73°29'56"E ~ 104°40'2"E) (Fig 1), with a total boundary length of 11,745.96 km and an area of 2,542.23×10³ km² [30]. The region is primarily composed of plateaus, mountains, basins, and valleys, characterized by significant topographical relief, high altitudes, and complex mountain ranges, with extreme climatic conditions. Additionally, the Qinghai-Tibet Plateau is influenced by monsoon and plateau climates, resulting in significant temperature differences and uneven spatial distribution of precipitation. As an important carbon storage region, the Qinghai-Tibet Plateau has vast underground and vegetation carbon reservoirs, significantly impacting the global carbon cycle. On the one hand, the rich vegetation and soil on the plateau are conducive to carbon fixation and storage, while the cold and dry climate conditions help slow the decomposition rate of organic carbon, promoting long-term carbon storage. On the other hand, the Qinghai-Tibet Plateau, as one of the world's major absorbers of atmospheric carbon dioxide ($CO_2$), absorbs large amounts of $CO_2$ through its extensive ecosystems such as grasslands and wetlands via photosynthesis, converting it into organic carbon and thereby mitigating the pace of global climate change. Therefore, the unique ecosystem of the Qinghai-Tibet Plateau has a crucial impact on global climate change and the carbon cycle.

### 2.2 Datasets

In selecting driving factors, two aspects were considered primarily: On the one hand, due to the Qinghai-Tibet Plateau ecosystem's high fragility and susceptibility to climate change (including rising temperatures, precipitation changes, and increasing aridity) and human activities (such as overgrazing and industrial development), it was deemed pertinent to select drivers related to ecosystem health—such as soil properties and vegetation cover—to assist researchers in identifying the principal factors influencing ecological restoration. The ecosystem services of the Qinghai-Tibet Plateau, including water conservation, soil retention, and ecotourism, are vital for the sustainable development of the region. The incorporation of variables pertaining to ecosystem services, such as the proximity to railways and roads, was undertaken with the objective of conducting a comprehensive assessment of the enhancement of ecological service value when evaluating the cost-effectiveness of restoration projects. This

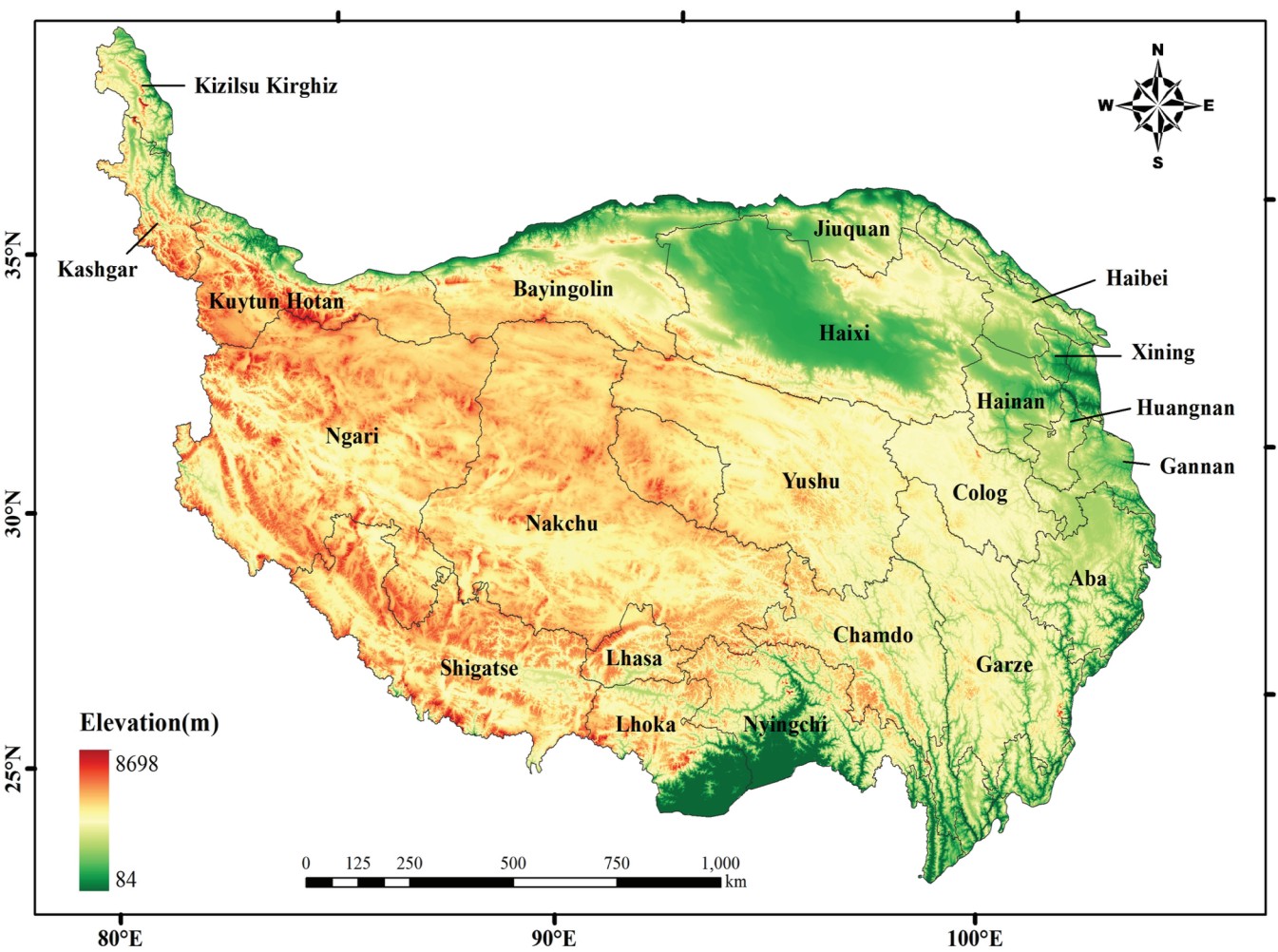

**Fig 1. Overview of the study area.**

approach provides scientific support for restoration efforts. In consideration of the region's complex terrain, with significant slope variability, which amplifies the impact of slope aspects, topographic factors such as slope and aspect were also included in the analysis. On the other hand, we referred to relevant previous studies to supplement and refine the selection of driving factors [31–34]. Fourteen driving factors were selected for the study of land use type and carbon storage changes (Table 1).

The land use and land cover(LULC) data from 2000 to 2020 were obtained from the Zenodo platform, provided by Wuhan University, with a spatial resolution of 30 meters. This dataset details land types, including nine categories: cropland, forest land, shrubland, grassland, water bodies, snow/ice, unused land, and built-up areas. In this study, the land use types were reclassified into six categories according to the Chinese Current Land Use Classification System, where shrubland and forest land were combined into forest land, water bodies, snow/ice were combined into water bodies, and wetlands were classified as unused land. The boundary map of the Tibetan Plateau is from the National Tibetan Plateau Data Center [35].

**Table 1. Datasets, indicators, and sources.**

| Data type | Data Name | Year | Resolution | Data Source |
|---|---|---|---|---|
| Land Use Data | LULC | 2000–2020 | 30 m | https://zenodo.org/records/8176941 |
| Carbon Density Data | CD | – | – | http://www.cnern.org.cn/ [36] |
| Natural Factors | Temp | 2000–2020 | 1 km | |
| | Pre | 2000–2020 | 1 km | https://data.tpdc.ac.cn/home |
| | AI | 2020 | 1 km | |
| | NDVI | 2000–2020 | 1 km | https://search.earthdata.nasa.gov/search |
| | Fs | 2017 | 1 km | |
| | St | – | 1 km | https://www.resdc.cn/ |
| Topographic Factors | DEM | 2020 | 90 m | |
| | Slop | 2020 | 90 m | http://srtm.csi.cgiar.org/srtmdata |
| | Aspect | 2020 | 90 m | |
| Social-Economic Factors | POP | 2010, 2020 | 1 km | https://landscan.ornl.gov |
| | GDP | 2010, 2019 | 1 km | https://doi.org/10.6084/m9.figshare.17004523.v1 |
| Distance Factors | Disrd | 2021 | 1 km | |
| | Disr | 2021 | 1 km | https://www.openstreetmap.org/ |
| | Disw | 2021 | 1 km | |

DEM data were obtained from the SRTM elevation data, jointly measured by NASA and the National Imagery and Mapping Agency (NIMA). Based on the DEM data, slope and aspect data were extracted using ArcGIS 10.8. Annual mean temperature(Temp) and precipitation(Pre) data were calculated using monthly precipitation raster data and monthly temperature raster data provided by the National Tibetan Plateau Data Center. Road data were provided by OpenStreetMap, and the Euclidean distance analysis tool in ArcGIS 10.8 was used to calculate the distances from each pixel to roads(Disrd), water bodies(Disw), and railways(Disr). Since carbon storage changes with land use type through the biochemical processes of ecosystems, this study used land use type data and carbon density data to calculate the carbon storage of the Qinghai-Tibet Plateau.

## 2.3 Research methods

**2.3.1 Land development scenarios** In this study, three development scenarios were set: "Natural Development Scenario(ND)", "Cropland Protection Scenario(CP)", and "Ecological Protection Scenario(EP)". In the natural development scenario, land use demand is not subject to human intervention, consistent with the land change patterns observed on the Qinghai-Tibet Plateau from 2010 to 2020. The Cropland Protection Scenario prioritizes agriculture by restricting non-agricultural land conversion, mitigating cropland reduction, and ensuring economic development. In this scenario, by assigning field weights, the probability of converting farmland to built-up areas decreases by 70% while reducing conversion to grassland, forest land, and water bodies by 40%. Conversely, the probability of converting unused land to farmland increases by 50%. The Ecological Protection Scenario prioritizes forest land, grassland, and water bodies as ecological land, restricting built-up area expansion. In this scenario, by setting field weights, the probability of converting cropland to forest land and grassland increases by 20%, while unused land has a 25% higher probability of conversion to forest land and grassland as reserve land resources [37]. The land use type transition matrices for the three scenarios are shown in Table 2.

**2.3.2 PLUS model** The PLUS (Patch-generating Land Use Simulation) model is a patch-generating land use simulation model that is essentially a type of cellular automata model. It uses the Markov model to predict land use demand and, combined with LEAS (Land Expansion Analysis Strategy), integrates multiple random seeds with cellular automata (CA),

**Table 2. Land use type transitions under three different scenarios.**

| Development Scenario | Land Use Type | Cropland | Forest Land | Grassland | Water Bodies | Unused Land | Built-up Areas |
|---|---|---|---|---|---|---|---|
| Natural Development | Cropland | 1 | 1 | 1 | 1 | 1 | 1 |
| | Forest Land | 1 | 1 | 1 | 1 | 1 | 1 |
| | Grassland | 1 | 1 | 1 | 1 | 1 | 1 |
| | Water Bodies | 1 | 1 | 1 | 1 | 1 | 0 |
| | Unused Land | 1 | 1 | 1 | 1 | 1 | 1 |
| | Built-up Areas | 0 | 0 | 0 | 0 | 0 | 1 |
| Cropland Protection | Cropland | 1 | 0 | 0 | 0 | 0 | 0 |
| | Forest Land | 1 | 1 | 1 | 1 | 1 | 1 |
| | Grassland | 1 | 1 | 1 | 1 | 1 | 1 |
| | Water Bodies | 1 | 1 | 1 | 1 | 1 | 0 |
| | Unused Land | 1 | 1 | 1 | 1 | 1 | 1 |
| | Built-up Areas | 0 | 0 | 0 | 0 | 0 | 1 |
| Ecological Protection | Cropland | 1 | 1 | 1 | 1 | 1 | 1 |
| | Forest Land | 0 | 1 | 1 | 0 | 0 | 0 |
| | Grassland | 0 | 1 | 1 | 0 | 0 | 0 |
| | Water Bodies | 1 | 1 | 1 | 1 | 1 | 0 |
| | Unused Land | 1 | 1 | 1 | 1 | 1 | 1 |
| | Built-up Areas | 0 | 0 | 0 | 0 | 0 | 1 |

offering higher simulation accuracy. This is particularly important for ecologically diverse regions under multiple pressures, such as the Qinghai-Tibet Plateau. The model can reveal the driving mechanisms behind land use changes, supporting the formulation of more precise restoration strategies.

**a. Markov module**

The Markov module predicts future land use demand based on historical land use transition probability matrices [38], as described by the following equation:

$$S_{(t+1)} = P_t \times S_t \tag{1}$$

where $S_{(t+1)}$ represents the land use type at time $t + 1$; $P_t$ represents the land use transition probability matrix; and $S_t$ represents the land use type at time $t$.

**b. LEAS**

LEAS uses the Random Forest Classifier (RFC) to determine the process of land use type change. RFC is a classification method based on an ensemble of decision trees, extracting random samples from the original training data set. It can handle high-dimensional data while addressing multicollinearity among variables, converting land use transition rules into binary classification problems, and exploring the relationship between land use changes and driving factors. The transition probability is calculated using the following formula:

$$P_{i,k}^d(x) = \frac{\sum_{n=1}^{M} I\left(h_n(x) = d\right)}{M} \tag{2}$$

where $P_{i,k}^d(x)$ represents the probability that plot $i$ will be converted to land use type $k$; $d$ is a binary variable with a value of 1 indicating that the plot is predicted to change and 0 otherwise; $x$ represents a vector of factors driving land use change; $h_n(x)$ represents the prediction result of the $n$-th tree in the random forest for whether plot $i$ will be converted to type

$k$; $M$ is the total number of decision trees in the random forest; and $I$ is an indicator function, with a value of 1 if the $n$-th tree predicts that the plot will be converted to type $k$, and 0 otherwise.

**c. CARS**

The CARS model simulates and predicts the spatial distribution of land use types under different scenarios by embedding a land use/land cover change model within the traditional cellular automata framework. It effectively integrates "top-down" drivers at the macro level with "bottom-up" operational mechanisms at the micro level. In this framework, global land use demand influences local land use competition through an adaptive inertia mechanism, driving land use patterns to meet future demand changes. The CARS module combines random seed patch generation and a descending threshold to perform future land use simulations on the CA model. When the neighborhood effect of a land use type disappears, the module generates new "seeds" for each land use type, which are randomly initialized patches to guide the development of new land use types. Finally, using these seed patches, the PLUS model automatically generates simulated land use data.

**d. Model validation**

This study predicts land use for 2030 and 2040 based on the PLUS model. To validate the model's accuracy and effectiveness, the study simulates land use in 2020 on the QTP (Qinghai-Tibet Plateau) using land use data from 2000 and 2010, then verifies the simulation results with actual 2020 land use data. The consistency of the PLUS model is typically validated using the Kappa coefficient, which is widely applied in land use classification and simulation accuracy assessments [39,40]. The Kappa coefficient was used to measure the consistency between the simulation results and actual land use. When the Kappa coefficient is >0.75, it indicates high simulation accuracy. In this study, the Kappa coefficient reached 0.88, indicating that the PLUS model is reasonably applicable to the Qinghai-Tibet Plateau.

**2.3.3 InVEST model** InVEST is an open-source decision-support model jointly developed by Stanford University, The Nature Conservancy, and the World Wildlife Fund (WWF) to help decision-makers assess the role and value of natural environments in economic development by quantifying the value of ecosystem services. Given the Qinghai-Tibet Plateau's role as a significant global carbon sink, the selection of the InVEST model for carbon storage assessment highlights its considerable ecological value and the critical role of carbon storage in climate regulation and ecosystem stability, enhancing our understanding of ecological potential and challenges under future scenarios. This study utilizes the carbon storage and sequestration module of the InVEST model, which automatically generates a spatial distribution map of carbon storage based on the regional distribution characteristics of different land use types and their respective carbon density data. Table 3 shows the sources of carbon density data, primarily from the Chinese Ecosystem Research Network Data Center and relevant literature. Carbon storage is primarily determined by four carbon pools:

*Aboveground Biomass Carbon ($C_{above}$):* Refers to the carbon stored in the aboveground parts of all plants in the ecosystem, such as trunks, bark, and leaves;

*Belowground Biomass Carbon ($C_{below}$):* Refers to the carbon stored in the belowground parts of plants, such as root systems;

*Soil Organic Carbon ($C_{soil}$):* Refers to the organic carbon distributed in organic and mineral soils;

*Dead Organic Matter Carbon ($C_{dead}$):* Refers to the carbon stored in dead biological tissues, mainly including fallen leaves, dead branches, fallen trees, etc.

**Table 3. Carbon density values for different land use types on the Qinghai-Tibet Plateau (t/ha).**

| Land Use Type | Aboveground Biomass | Belowground Biomass | Soil Biomass |
|---|---|---|---|
| Cropland | 3.73 | 8.2 | 50.38 |
| Forest Land | 43.1 | 17.6 | 204.77 |
| Grassland | 0.74 | 6.43 | 109.04 |
| Water Bodies | 0.3 | 0 | 0 |
| Unused Land | 0.92 | 0 | 26.45 |
| Built-up Areas | 0 | 0 | 0 |

The formula for calculating carbon storage is as follows:

$$C_i = C_{i\_above} + C_{i\_below} + C_{i\_soil} + C_{i\_dead} \tag{3}$$

$$C_{total} = \sum_{i=1}^{n} C_i \times S_i \tag{4}$$

where $C_i$ is the total carbon density (t/ha) of land use type $i$, $C_{total}$ is the total carbon storage (t), $S_i$ is the total area (ha) of land use type $i$, and $n$ is the number of land use types, where $n = 6$. Due to the difficulty in collecting data on dead organic matter carbon and its minimal impact on carbon storage [41], this study does not consider it.

**2.3.4 Method for analyzing influencing factors** *Ordinary Least Squares (OLS)*. OLS is a classical regression technique that calculates fixed regression coefficients for the entire dataset, ignoring any spatial heterogeneity. Therefore, we utilized OLS to reveal the relationship between carbon storage and driving factors across the entire Qinghai-Tibet Plateau. The formula for the OLS regression equation is as follows:

$$y_i = \beta_0 + \sum_{j=1}^{n} \beta_j x_{ij} + \epsilon \tag{5}$$

where $y_i$ represents the carbon storage in the $i$-th city; $\beta_0$ denotes the intercept; $x_{ij}$ represents the value of the $j$-th independent variable in the $i$-th city; $\beta_j$ is the regression coefficient for the $j$-th independent variable; $\epsilon$ is the error term; and $n$ is the number of explanatory variables.

*Geographically Weighted Regression (GWR)*. GWR is a spatial statistical method used to analyze the influence of geographic location on regression coefficients. Unlike traditional linear regression models, GWR allows for varying regression coefficients across geographic space, capturing the local effects of geographic features on variable relationships. GWR is employed to explain spatial heterogeneity between dependent variables and independent driving factors [36]. In this study, we used the carbon storage and driving factors of each county within the Qinghai-Tibet Plateau in 2020 to train the GWR model, discussing analysis results. The general formula for the GWR model is:

$$y_i = \beta_0(u_i, v_i) + \sum_m \beta_m(u_i, v_i) x_{im} + \epsilon_i \tag{6}$$

where the dependent variable is $y_i$, the parameter value of the independent variable $x_m$ at spatial location $i$ is $x_{im}$, the coordinates of the spatial location are $(u_i, v_i)$, the intercept of the regression equation is represented by $\beta_0(u_i, v_i)$, $m$ represents the number of independent variables in the model, and $\epsilon_i$ is the error term.

When studying a vast and complex region like the Qinghai-Tibet Plateau, OLS can help researchers understand the average impact of major driving factors across the entire area, laying a foundation for preliminary analysis and deeper regional studies. On the other hand, GWR allows regression coefficients to vary across geographic space, capturing the local effects of geographic features on variable relationships. Combining both methods is particularly suitable for regions like the Qinghai-Tibet Plateau, where geographic and climatic conditions are complex and regional differences are significant. This approach not only provides trends and overall relationships at a global scale but also reveals how different driving factors impact carbon storage and ecosystem services in various regions, offering more targeted insights for refined management in diverse areas.

## 3 Experimental results

### 3.1 Analysis of land type changes on the Qinghai-Tibet Plateau from 2000 to 2020

In 2020, the areas of grassland, unused land, forest land, water bodies, cropland, and built-up areas on the Qinghai-Tibet Plateau were approximately 1,618,695 square kilometers, 614,232 square kilometers, 249,343 square kilometers, 117,008 square kilometers, 9,677 square kilometers, and 167 square kilometers, respectively. In terms of spatial distribution, grassland is mainly located in the central and eastern regions; unused land is concentrated in the western and northern border areas; forest land is concentrated in the southeastern part; and water bodies, cropland, and built-up areas, which cover relatively smaller areas, are primarily found in the western, eastern, and southeastern regions. Grassland is the dominant land use type on the Qinghai-Tibet Plateau, accounting for approximately 62.04% of the total area, followed by unused land, which accounts for about 23.54%. Built-up areas have the smallest share, accounting for about 0.006% (Fig 2). From 2000 to 2020, the areas of unused land and cropland showed a decreasing trend, decreasing by 16,147 square kilometers and 1,202 square kilometers, respectively (Table 4). The proportions of unused land and cropland decreased from 24.16% and 0.42% to 23.54% and 0.37%, respectively. Water bodies saw the largest increase in area, with an increase of 9,422 square kilometers, followed by forest land (7,148 square kilometers) and grassland (699 square kilometers). Although built-up areas experienced a high rate of change, the increase in area was small, at only 80 square kilometers. From 2000 to 2020, the total area change on the Qinghai-Tibet Plateau was 177,144 square kilometers, accounting for 6.79% of the total area. The land type transition matrix for the Qinghai-Tibet Plateau is shown in Table 5.

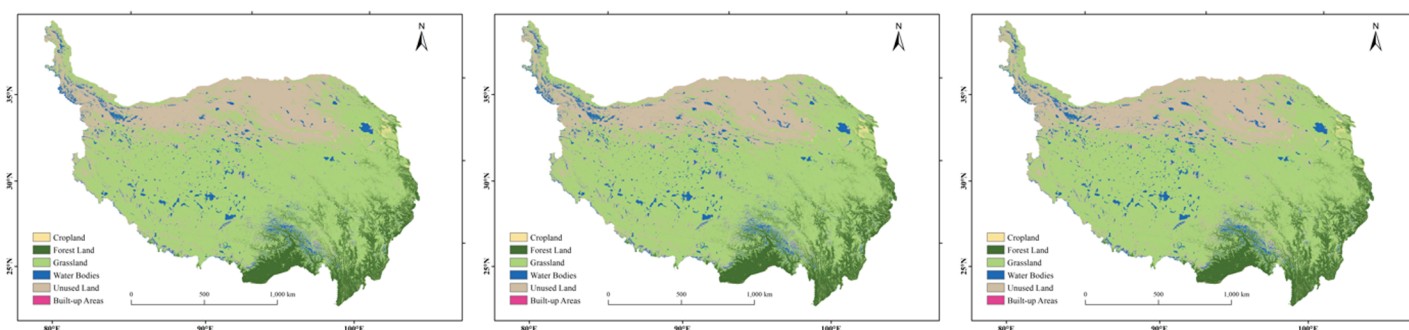

**Fig 2. Land use distribution on the Qinghai-Tibet Plateau in 2000, 2010, and 2020.**

**Table 4. Changes and dynamics of land use types on the Qinghai-Tibet Plateau from 2000 to 2020.**

| Land Type | Area/km² | | | Area Changes and Dynamics from 2000 to 2020 | |
|---|---|---|---|---|---|
| | 2000 | 2010 | 2020 | Area/km² | Dynamics% |
| Cropland | 10,879 | 10,267 | 9,677 | −1,202 | −11.04% |
| Forest Land | 242,195 | 246,982 | 249,343 | 7,148 | 2.95% |
| Grassland | 1,617,992 | 1,617,892 | 1,618,695 | 703 | 0.04% |
| Water Bodies | 107,586 | 124,591 | 117,008 | 9,422 | 8.76% |
| Unused Land | 630,379 | 609,255 | 614,232 | −16,147 | −2.56% |
| Built-up Areas | 87 | 135 | 167 | 80 | 91.95% |

## 3.2 Analysis of carbon storage changes on the Qinghai-Tibet Plateau from 2000 to 2020

**3.2.1 Temporal variation characteristics of Carbon storage**   This study utilized the carbon storage and carbon sequestration modules of the InVEST model to calculate the carbon storage content on the Tibetan Plateau from 2000 to 2020, as shown in Fig 3(A). During this period, the total carbon storage on the Tibetan Plateau generally exhibited a fluctuating upward trend. By 2020, the total carbon storage had increased by $1.4648 \times 10^8$ t compared to 2000. For example, the total carbon storage in 2000, 2010, and 2020 was $2.7029 \times 10^{10}$ t, $2.7093 \times 10^{10}$ t, and $2.7175 \times 10^{10}$ t, respectively. Among these, soil organic carbon is the primary type of carbon storage, with values of $2.4324 \times 10^{10}$ t, $2.4362 \times 10^{10}$ t, and $2.4429 \times 10^{10}$ t, accounting for 89.99%, 89.92%, and 89.90% of the total carbon storage, respectively. Analyzing the overall changes in carbon storage on the Tibetan Plateau, it can be observed that due to increases in aboveground biomass, belowground biomass, and soil organic carbon to varying degrees, the total carbon storage has shown an overall upward trend, as illustrated in Fig 3(B).In addition, the carbon storage content of various land use types over the three years is shown in Table 6.

During the period from 2000 to 2020, the primary source of carbon storage on the Tibetan Plateau was grassland. The carbon storage in grassland in 2000, 2010, and 2020 was $1.8803 \times 10^{10}$ t, $1.8802 \times 10^{10}$ t, and $1.8811 \times 10^{10}$ t, respectively, accounting for 69.57%, 69.40%, and 69.22% of the total carbon storage. The next major source was forest land, with carbon storage of $6.4296 \times 10^9$ t, $6.5566 \times 10^9$ t, and $6.6193 \times 10^9$ t, accounting for 23.79%, 24.20%, and 24.36% of the total carbon storage, respectively. Among other land use types, the carbon storage in cropland and unused land, which accounted for 0.25% and 6.38% of the total carbon storage in 2000, showed significant declines, with a year-on-year decrease of 11.05% and 2.56% in 2020, respectively. Overall, the carbon storage in water bodies showed a trend of increasing and then decreasing, with a decrease of 8.76% in 2020 compared to 2000.

**Table 5. Land type transfer matrix on the Qinghai-Tibet Plateau from 2000 to 2020.**

| | Transfer Amount from 2000 to 2020 (km²) | | | | | | Total |
|---|---|---|---|---|---|---|---|
| | Grassland | Cropland | Built-up Areas | Forest Land | Water Bodies | Unused Land | |
| Grassland | 1,545,289 | 1,523 | 31 | 10,553 | 5,746 | 54,850 | 1,617,992 |
| Cropland | 3,167 | 7,167 | 29 | 415 | 85 | 16 | 10,879 |
| Built-up Areas | – | – | 81 | – | 6 | – | 87 |
| Forest Land | 2,957 | 917 | 11 | 238,301 | 7 | 2 | 242,195 |
| Water Bodies | 1,925 | 6 | 9 | 50 | 93,684 | 11,912 | 107,586 |
| Unused Land | 65,353 | 64 | 6 | 24 | 17,480 | 547,452 | 630,379 |
| Total | 1,618,691 | 9,677 | 167 | 249,343 | 117,008 | 614,232 | 2,609,118 |

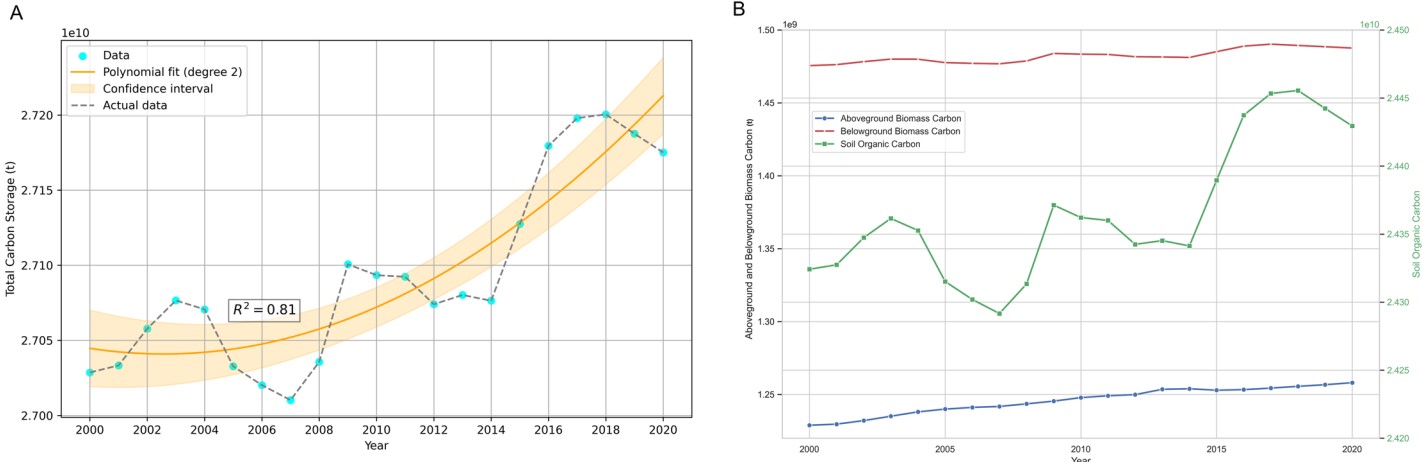

**Fig 3. Temporal dynamics and composition of carbon storage on the Qinghai-Tibet Plateau (2000–2020).** (A) Changes in total carbon storage on the Qinghai-Tibet Plateau from 2000 to 2020. (B) Carbon components over the years.

**Table 6. Carbon storage for each land use type in 2000, 2010, and 2020 (t).**

| Year | Cropland | Forest Land | Grassland | Water Bodies | Unused Land | Built-up Areas | Total |
|---|---|---|---|---|---|---|---|
| 2000 | 67,787,049 | 6,429,550,665 | 18,802,731,516 | 3,227,580 | 1,725,347,323 | 0 | 27,028,644,133 |
| 2010 | 63,973,677 | 6,556,631,154 | 18,801,522,932 | 3,737,730 | 1,667,530,935 | 0 | 27,093,396,428 |
| 2020 | 60,297,387 | 6,619,308,621 | 18,810,854,595 | 3,510,240 | 1,681,152,984 | 0 | 27,175,123,827 |

**3.2.2 Spatial variation characteristics of carbon storage** The distribution of carbon storage on the Tibetan Plateau shows a decreasing trend from southeast to northwest spatially (Fig 4). The region with the highest carbon storage is Nagqu, located in the central part of the Tibetan Plateau, with carbon storage of $3.5604 \times 10^9$ t, $3.5307 \times 10^9$ t, and $3.5256 \times 10^9$ t in 2000, 2010, and 2020, respectively. The next highest is the Ali region, with carbon storage of $3.1076 \times 10^9$ t, $3.0778 \times 10^9$ t, and $3.1607 \times 10^9$ t. The region with the lowest carbon storage is Xining, with $8.8388 \times 10^7$ t, $8.8429 \times 10^7$ t, and $9.1091 \times 10^7$ t, respectively. During the period from 2000 to 2020, the Ali region experienced the largest increase in carbon storage, with a decrease of $2.9770 \times 10^7$ t from 2000 to 2010 and an increase of $8.2896 \times 10^7$ t from 2010 to 2020, resulting in a net increase of $5.3111 \times 10^7$ t. Throughout the study period, Yushu Tibetan Autonomous Prefecture saw a continuous decrease in carbon storage, with reductions of $1.9212 \times 10^7$ t from 2000 to 2010 and $3.3778 \times 10^7$ t from 2010 to 2020, totaling a decrease of $5.2990 \times 10^7$ t. Conversely, Garze Tibetan Autonomous Prefecture experienced a continuous increase in carbon storage by $2.4513 \times 10^7$ t, with increases of $1.8731 \times 10^7$ t from 2000 to 2010 and $5.7821 \times 10^6$ t from 2010 to 2020. From 2000 to 2020, regions such as Nyingchi, Kashgar, Aba Tibetan and Qiang Autonomous Prefecture exhibited an increasing trend in carbon storage, with net increases of $1.6281 \times 10^7$ t, $9.7337 \times 10^6$ t, and $7.2048 \times 10^6$ t, respectively. In contrast, regions such as Hainan Tibetan Autonomous Prefecture, Haibei Tibetan Autonomous Prefecture, and Kizilsu Kirghiz Autonomous Prefecture showed a decreasing trend, with net reductions of $5.2483 \times 10^6$ t, $5.1638 \times 10^6$ t, and $3.7246 \times 10^6$ t, respectively. Due to the influence of vegetation cover, land use types, and climatic conditions, southeastern regions such as Nyingchi and Garze Tibetan Autonomous Prefecture have higher carbon storage due to their high vegetation cover, predominantly consisting of forest land and grassland, and a warm, moist climate conducive to plant growth, which promotes carbon fixation

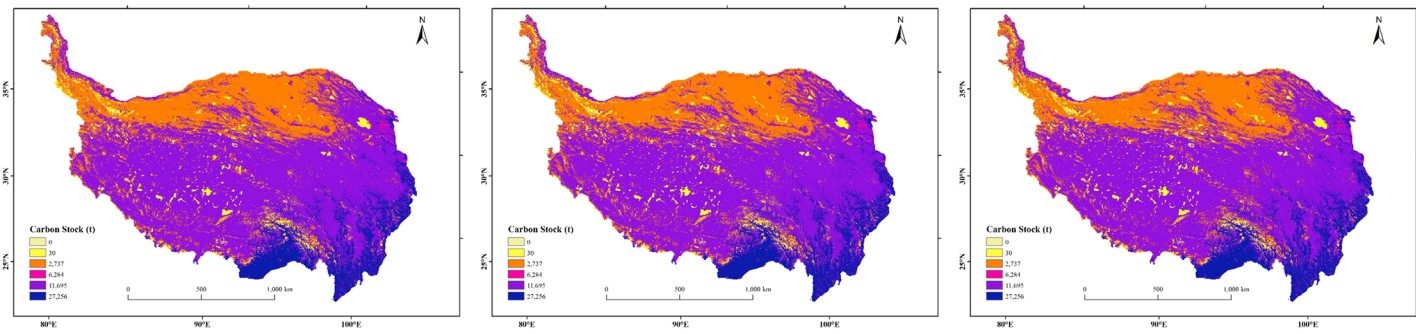

**Fig 4. Spatial distribution of carbon storage on the Qinghai-Tibet Plateau in 2000, 2010, and 2020.**

and storage. Nagqu, located in the central Tibetan Plateau, although having a colder climate, is primarily covered by alpine meadows, which also contribute to significant carbon storage. However, the northwestern regions, due to their dry and cold climate, sparse vegetation, and predominantly desert landscapes, have lower carbon storage.

The regional distribution of changes in carbon storage on the Tibetan Plateau from 2000 to 2020 is shown in Fig 5. Most areas of the Tibetan Plateau experienced stable changes in carbon storage, with regions showing no significant change covering an area of 2,430,564 km$^2$, accounting for 93.16% of the total area3.2.3 ($p$ <0.05). In contrast, 3.75% of the area experienced significant increases in carbon storage, scattered across regions such as Jiuquan, Kashgar, Kizilsu Kirghiz Autonomous Prefecture, and Ali, with a total area of 97,894 km$^2$ (*slope* >0, $p \leq$0.05). Among these, Jiuquan had the most significant increase, with the area of significant carbon storage increase accounting for 10.44% of the total area of the region. Conversely, 3.09% of the area saw significant decreases in carbon storage, scattered across Kizilsu Kirghiz Autonomous Prefecture, Kashgar, Hotan, and Bayingolin Mongol Autonomous Prefecture, with a total area of 80,549 km$^2$ (*slope* <0, $p \leq$0.05). Notably, Kizilsu Kirghiz Autonomous Prefecture experienced the most significant decrease, with the area of significant carbon storage decrease accounting for 7.54% of the total area of the region (*slope* indicates the change rate and $p$ is the significance test value).

**3.2.3 Analysis of factors influencing carbon storage** *Global Analysis of Factors Influencing Carbon Storage* This study employs the OLS regression method to investigate the impact of factors such as temperature, precipitation, and aridity index on carbon storage in the Tibetan Plateau. Among the 14 driving factors examined, frozen soil type (Fs) and soil type (St) are categorical data, so these two driving factors are not included in the OLS and GWR models. The results (Table 7) indicate that the constructed OLS model can effectively fit the relationship between carbon storage and the driving factors, with an $R^2$ of 0.55. Six factors, namely DEM, average annual temperature, distance from railways, aridity index, distance from roads, and NDVI exhibited high significance. Among these, DEM, average annual temperature, distance from railways, distance from roads, and NDVI acted as positive explanatory variables, whereas the aridity index was a negative explanatory variable. The regression coefficients for the positive explanatory variables DEM, average annual temperature, distance from railways, distance from roads, and NDVI are $1.66\times^{-1}$, $3.93\times10^{-1}$, $1.27\times10^{-1}$, $6.67\times10^{-2}$, and $5.71 \times 10^{-1}$, respectively. The regression coefficient for the negative explanatory variable aridity index is $-1.06\times10^{-1}$.

*Analysis of Local Carbon Storage Influencing Factors* This paper uses the GWR model to study the spatial heterogeneity of the relationship between carbon storage and driving factors

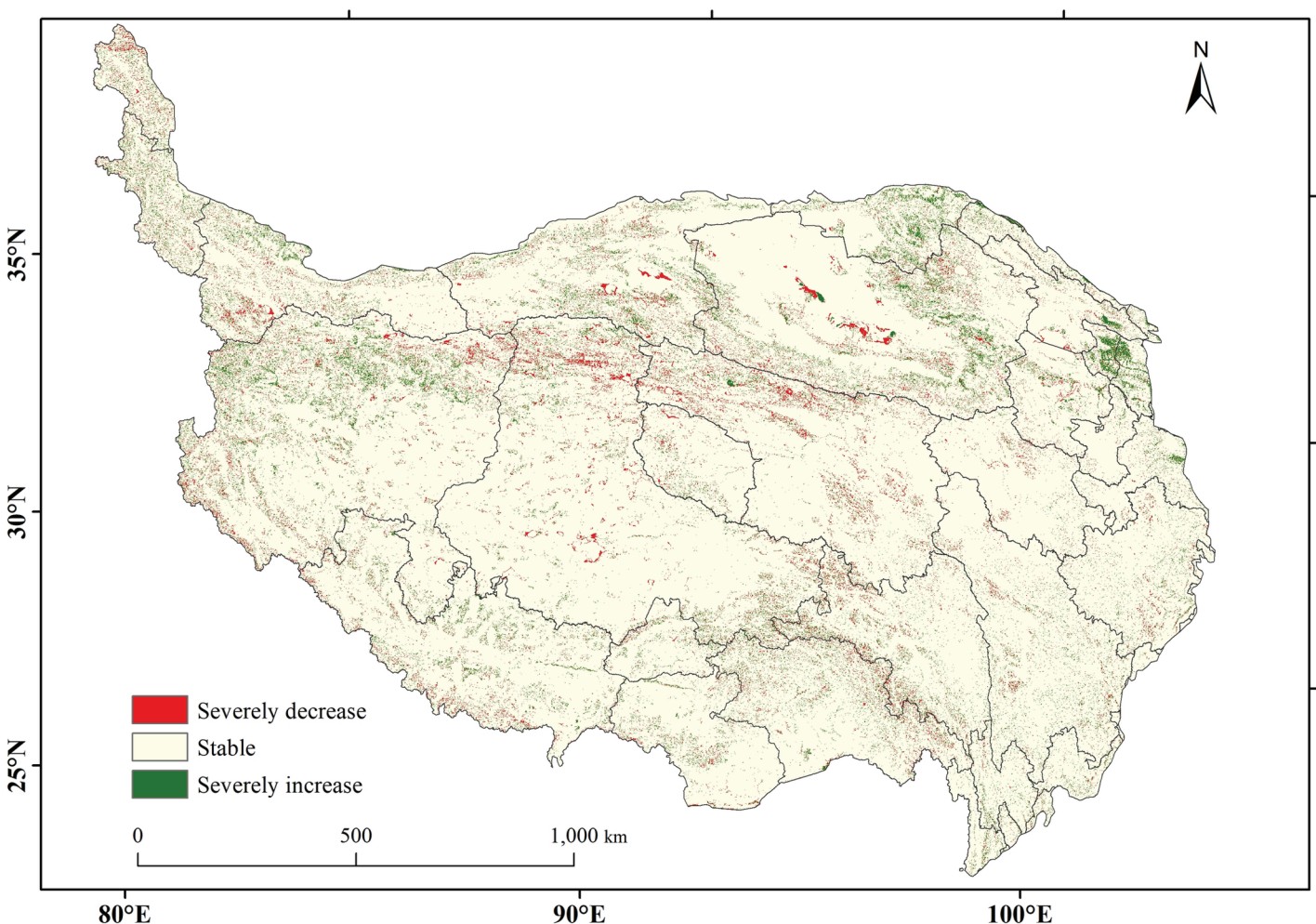

**Fig 5. Spatial changes in carbon storage on the Qinghai-Tibet Plateau.**

across various cities in the Tibetan Plateau in 2020, with an $R^2$ of 0.74. The calculation results are shown in Fig 6.

Among the remaining twelve factors, DEM, distance from railways, and distance from roads exhibit a spatial positive correlation with carbon storage. DEM has the most significant impact on carbon storage in the northwestern region of the Tibetan Plateau (Fig 6(a)), while the regions with significant effects from distance to railways and distance to roads are mainly located in the southeastern part of the Tibetan Plateau (Fig 6(j) and 6(k)).

Eight driving factors—slope, aspect, aridity index, gross domestic product (GDP), population (POP), average annual temperature, distance from water bodies, and NDVI—show different polarities in different regions. GDP exhibits a positive correlation with carbon storage in the central and western regions of the Tibetan Plateau, while showing a negative correlation in the northwest (Fig 6(e)); population shows the opposite pattern to GDP (Fig 6(f)); average annual temperature shows a positive correlation in small areas in the southeastern and central-northern parts of the Tibetan Plateau, while being negatively correlated in other regions (Fig 6(h)); aspect, aridity index, distance from water bodies, and NDVI show a transition from positive to negative correlation with carbon storage from east to north (Fig 6(c),

**Table 7. Results of factors influencing carbon storage on the Qinghai-Tibet Plateau based on OLS.**

|  | Estimate | Std. Error | t value | Pr(>|t|) |
|---|---|---|---|---|
| (Intercept) | $-7.10 \times 10^{-17}$ | $1.91 \times 10^{-2}$ | $-4.12 \times 10^{-15}$ | $1.00 \times 10^{0}$ |
| DEM | $1.66 \times 10^{-1}$ | $3.55 \times 10^{-2}$ | $4.67 \times 10^{0}$ | $5.42 \times 10^{-3}**$ |
| Slop | $1.56 \times 10^{-2}$ | $2.66 \times 10^{-2}$ | $5.97 \times 10^{-1}$ | $3.86 \times 10^{-1}$ |
| Aspect | $1.36 \times 10^{-2}$ | $2.66 \times 10^{-2}$ | $5.04 \times 10^{-1}$ | $4.69 \times 10^{-1}$ |
| AI | $-1.06 \times 10^{-1}$ | $2.54 \times 10^{-2}$ | $-4.18 \times 10^{0}$ | $9.28 \times 10^{-4}***$ |
| GDP | $5.58 \times 10^{-3}$ | $2.55 \times 10^{-2}$ | $-1.44 \times 10^{-1}$ | $4.06 \times 10^{-1}$ |
| POP | $-4.72 \times 10^{-2}$ | $2.59 \times 10^{-2}$ | $-1.82 \times 10^{0}$ | $2.88 \times 10^{-1}$ |
| Pre | $-7.29 \times 10^{-2}$ | $2.83 \times 10^{-2}$ | $-2.58 \times 10^{0}$ | $8.48 \times 10^{-2}$ |
| Temp | $3.93 \times 10^{-1}$ | $3.60 \times 10^{-2}$ | $1.09 \times 10^{1}$ | $8.47 \times 10^{-21}***$ |
| Disw | $1.96 \times 10^{-2}$ | $2.18 \times 10^{-2}$ | $8.99 \times 10^{-1}$ | $4.26 \times 10^{-1}$ |
| Disr | $1.27 \times 10^{-1}$ | $2.09 \times 10^{-2}$ | $6.09 \times 10^{0}$ | $2.62 \times 10^{-5}***$ |
| Disrd | $6.67 \times 10^{-2}$ | $2.14 \times 10^{-2}$ | $3.11 \times 10^{0}$ | $1.12 \times 10^{-2}**$ |
| NDVI | $5.71 \times 10^{-1}$ | $3.48 \times 10^{-2}$ | $1.64 \times 10^{1}$ | $1.57 \times 10^{-42}***$ |
| $R^2$ | $5.50 \times 10^{-1}$ |  | $F_{statistic}$ | $1.26 \times 10^{2}$ |
| $Adj.R^2$ | $5.50 \times 10^{-1}$ |  | $F_{p-value}$ | $7.91 \times 10^{-171}$ |

\* Significant at the 0.05 level; \*\* Significant at the 0.01 level; \*\*\* Significant at the 0.001 level.

6(d), 6(i), 6(l)), while slope shows the opposite trend (Fig 6(b)).In contrast, average annual precipitation shows a negative correlation with carbon storage, especially in the northwestern region of the Tibetan Plateau, where it is most significant (Fig 6(g)).

## 3.3 Prediction and analysis of carbon storage in land use types on the Qinghai-Tibet Plateau

**3.3.1 Land use change analysis under multiple scenarios** Based on the PLUS model, this study predicts the area of land use types on the Tibetan Plateau for 2030 and 2040 under three different scenarios (Table 8), their spatiotemporal changes (Fig 7), and the corresponding land use type transitions (Fig 8).

Under the natural development scenario, from 2020 to 2040, the overall change trends for forest land and grassland are similar to those from 2000 to 2020. The area of forest land and grassland is projected to increase by 4,597 km$^2$ and 1,927 km$^2$, respectively, with growth rates of 1.84% and 0.12%. The increase in forest land is primarily concentrated in the southeastern part of the Tibetan Plateau, including the Ganzi Tibetan Autonomous Prefecture (1,242 km$^2$) and Changdu (907 km$^2$). The increase in grassland is mainly in the southwestern part of the Tibetan Plateau, including Shigatse (620 km$^2$) and Ali (2,253 km$^2$). The trend of unused land shows a significant slowdown, with a net increase of 5,382 km$^2$ (0.88%). Built-up areas, with a smaller base, increased by 60 km$^2$, but this represents a 35.93% growth. The growth in unused land is mainly concentrated in the southeastern part of the Tibetan Plateau, including Changdu (2,070 km$^2$) and the northeastern part of Haixi Mongol and Tibetan Autonomous Prefecture (1,624 km$^2$). The growth in built-up areas is mainly concentrated in Shannan and Nyingchi (35 km$^2$ and 16 km$^2$). Cropland and water bodies are projected to decrease, with net reductions of 727 km$^2$ and 11,239 km$^2$, respectively (decreases of 7.51% and 9.61%). The reduction in cropland is mainly concentrated in Aba Tibetan and Qiang Autonomous Prefecture in the east (-270 km$^2$) and Deqin Tibetan Autonomous Prefecture in the southeast (-190 km$^2$). The reduction in water bodies is concentrated in Kashgar (-2,303 km$^2$), Hotan (-2,281 km$^2$), and Changdu (-1,122 km$^2$).

Under the cropland protection scenario, strict restrictions were placed on the conversion of cropland to other land use types, resulting in a significant increase in cropland area.

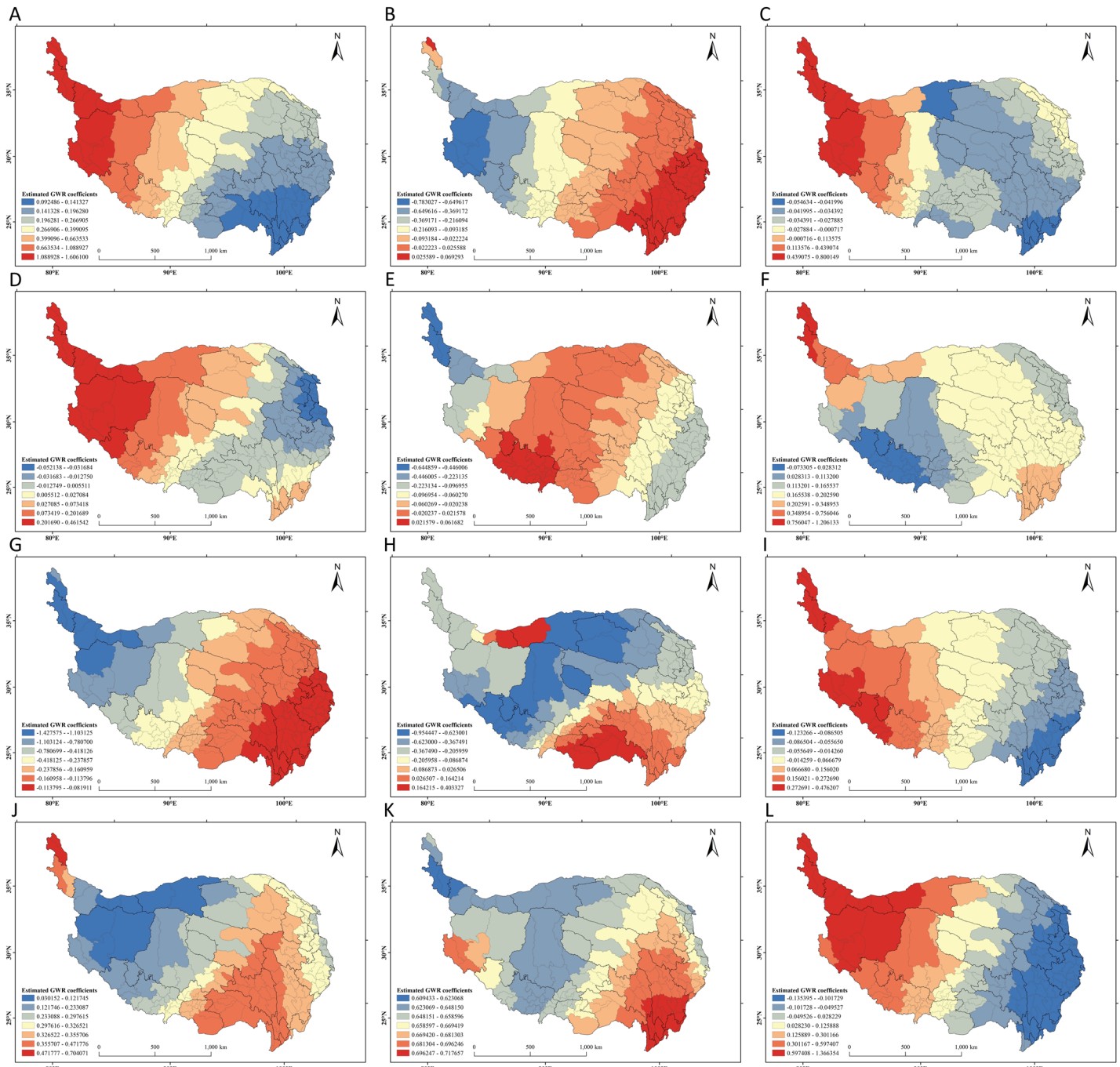

**Fig 6. Geographically weighted regression analysis of driving factors.** (A) DEM; (B) Slop; (C) Aspect; (D) AI; (E) GDP; (F) POP; (G) Pre; (H) Temp; (I) Disw; (J) Disr; (K) Disrd; (L) NDVI.

Compared to 2020, the cropland area is projected to net increase by 621 km² (6.42%) in 2030 and 1,149 km² (11.81%) in 2040. From 2020 to 2040, the increase in cropland is mainly concentrated in the central and southern parts, including Shannan and Nyingchi (384 km² and 222 km², respectively). Forest land area is also projected to show a certain degree of increase.

**Table 8. Area of land use types on the Qinghai-Tibet Plateau under different scenarios in 2030 and 2040.**

| Land Use Type | Area of Land Use Types in 2030/km$^2$ | | | Area of Land Use Types in 2040/km$^2$ | | |
|---|---|---|---|---|---|---|
| | ND | CP | EP | ND | CP | EP |
| Cropland | 9,253 | 10,298 | 8,748 | 8,950 | 10,826 | 8,136 |
| Forest Land | 251,660 | 251,559 | 251,717 | 253,940 | 253,756 | 254,075 |
| Grassland | 1,619,632 | 1,618,717 | 1,633,137 | 1,620,622 | 1,619,013 | 1,645,693 |
| Water Bodies | 110,824 | 110,804 | 110,561 | 105,769 | 105,741 | 105,075 |
| Unused Land | 617,556 | 617,554 | 604,762 | 620,716 | 619,575 | 595,917 |
| Built-up Areas | 197 | 190 | 197 | 227 | 211 | 226 |

ND is the Natural Development Scenario, CP is the Cropland Protection Scenario, and EP is the Ecological Protection Scenario.

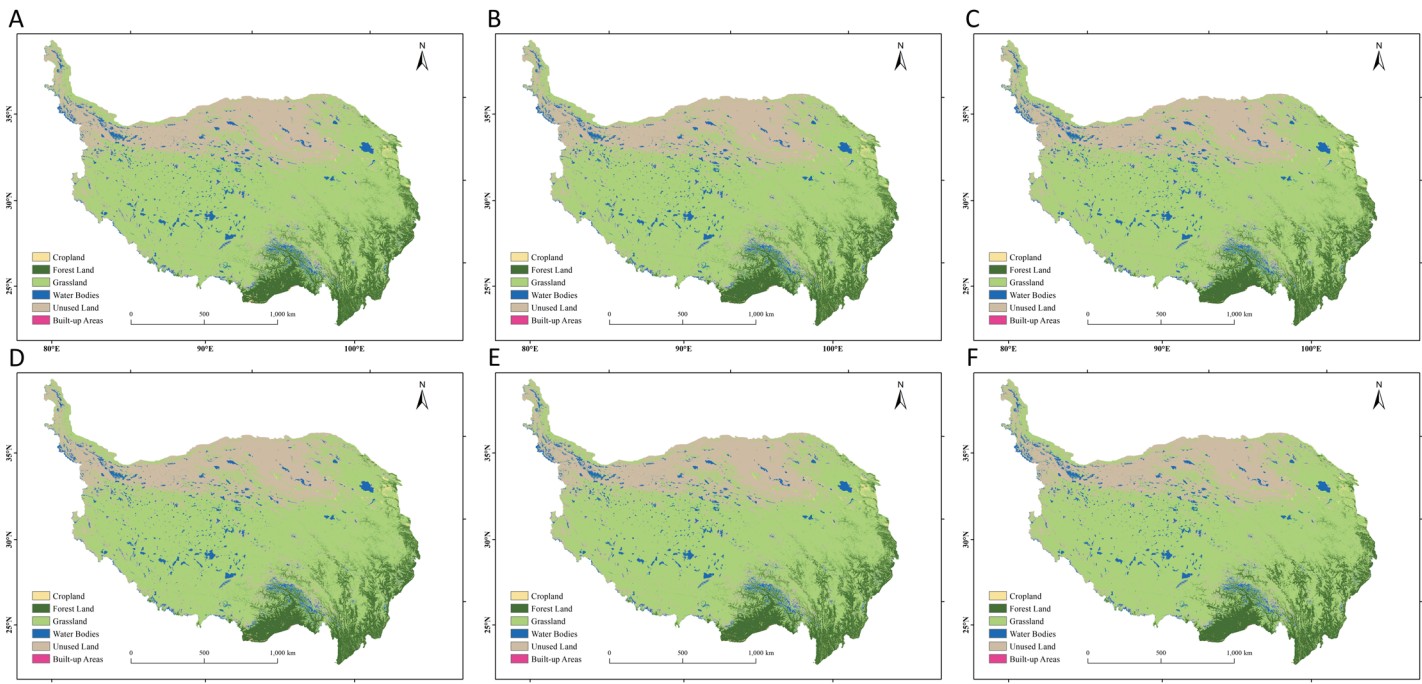

**Fig 7. Predicted distribution of land use types on the Qinghai-Tibet Plateau under different scenarios in 2030 and 2040.** (A) Natural Development Scenario in 2030; (B) Cropland Protection Scenario in 2030; (C) Ecological Protection Scenario in 2030; (D) Natural Development Scenario in 2040; (E) Cropland Protection Scenario in 2040; (F) Ecological Protection Scenario in 2040.

Compared to 2020, forest land is expected to increase by 2,216 km$^2$ (0.89%) in 2030 and 4,413 km$^2$ (1.77%) in 2040. From 2020 to 2040, the growth in forest land is primarily concentrated in the southeastern part of the Tibetan Plateau, including Ganzi Tibetan Autonomous Prefecture, Changdu, and Nyingchi, with increases of 1,255 km$^2$, 1,178 km$^2$, and 880 km$^2$, respectively. In contrast, the area of water bodies is projected to decrease significantly. Compared to 2020, water bodies are expected to decrease by 11,267 km$^2$ in 2040, representing a 9.63% reduction. The decrease in water bodies is concentrated in the southwestern part of the Tibetan Plateau, including Shigatse (–2,015 km$^2$), Ali (–1,988 km$^2$), and Nagqu (–1,818 km$^2$). Grassland and unused land show only minor fluctuations and are expected to remain relatively stable.

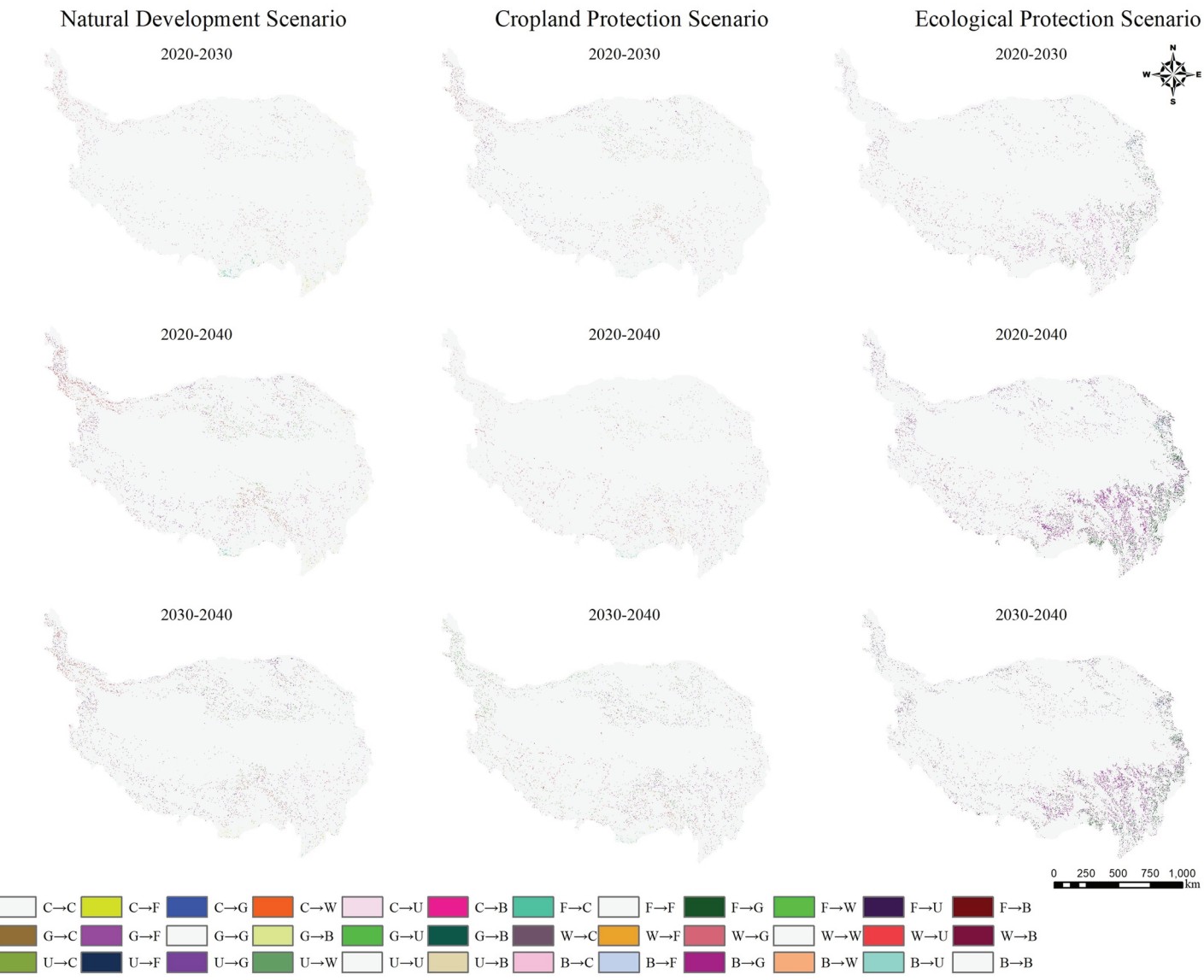

**Fig 8. Land use type transition distribution on the Tibetan Plateau under different scenarios from 2020 to 2040.** Land use type abbreviations are defined as follows: C: Cropland; F: Forest Land; G: Grassland; W: Water Bodies; U: Unused Land; B: Built-up Areas.

In the ecological protection scenario, compared to 2020, grassland is projected to net increase by 26,998 km$^2$ by 2040, representing a 1.67% increase. The growth is mainly concentrated in the Ali region (5,395 km$^2$) and Shigatse (5,153 km$^2$). The net increase in forest land area is 4,732 km$^2$, with a growth rate of 1.90%. The increase is mainly concentrated in Chamdo City (6,217 km$^2$) and Garzê Tibetan Autonomous Prefecture (2,240 km$^2$), while the decrease is mainly concentrated in Ngawa Tibetan and Qiang Autonomous Prefecture (−2,532 km$^2$) and Gannan Tibetan Autonomous Prefecture (−1,508 km$^2$). In contrast, unused land is projected to decrease by 18,315 km$^2$ compared to 2020, representing a 2.98% reduction. The decrease is mainly concentrated in the Ali region (−3,380 km$^2$), Shigatse (−2,973 km$^2$), Bayingol Mongol Autonomous Prefecture (−2,588 km$^2$), and Haixi Mongol and Tibetan Autonomous Prefecture (−2,314 km$^2$). Additionally, the area of water bodies is expected to

decrease by 11,933 km$^2$, a 10.20% reduction, with decreases primarily in Shigatse (−2,034 km$^2$), Ali region (−1,997 km$^2$), and Nyingchi (−1,924 km$^2$). Furthermore, cropland is projected to decrease by 1,541 km$^2$, representing a 15.92% reduction. The decrease is mainly concentrated in Xining (−481 km$^2$), Haidong (−398 km$^2$), and Haixi Mongol and Tibetan Autonomous Prefecture (−156 km$^2$). Built-up areas show little change, with a small increase of 59 km$^2$ (35.33%).

**3.3.2 Analysis of carbon storage changes under multiple scenarios** From 2000 to 2020, the carbon storage on the Qinghai-Tibet Plateau exhibited a generally fluctuating upward trend. Based on the InVEST model, this study forecasts the carbon storage for 2030 and 2040 under three different scenarios (Fig 9). In all three scenarios, the carbon storage in 2030 and 2040 shows an increasing trend compared to 2020. In the natural development scenario, the carbon stosrage increases by $7.87 \times 10^7$ t and $1.54 \times 10^8$ t in 2030 and 2040, respectively, mainly due to the rapid growth of forest land with high carbon density. In the cropland protection scenario, the carbon storage increases by $7.19 \times 10^7$ t and $1.42 \times 10^8$ t in 2030 and 2040, respectively. Overall, the increase in carbon storage is still primarily attributed to the expansion of forest land. In the ecological protection scenario, the carbon storage increases by $1.99 \times 10^8$ t and $3.79 \times 10^8$ t in 2030 and 2040, respectively. In this scenario, the substantial increase in the area of high carbon-density forest land and grassland leads to a rapid rise in overall carbon storage. In summary, carbon storage on the Qinghai-Tibet Plateau shows an increasing trend under all three scenarios: the ecological protection scenario has the fastest growth, followed by the natural development scenario, with the cropland protection scenario showing the slowest growth. The carbon storage changes from 2020 to 2040 under different scenarios are shown in Fig 10.

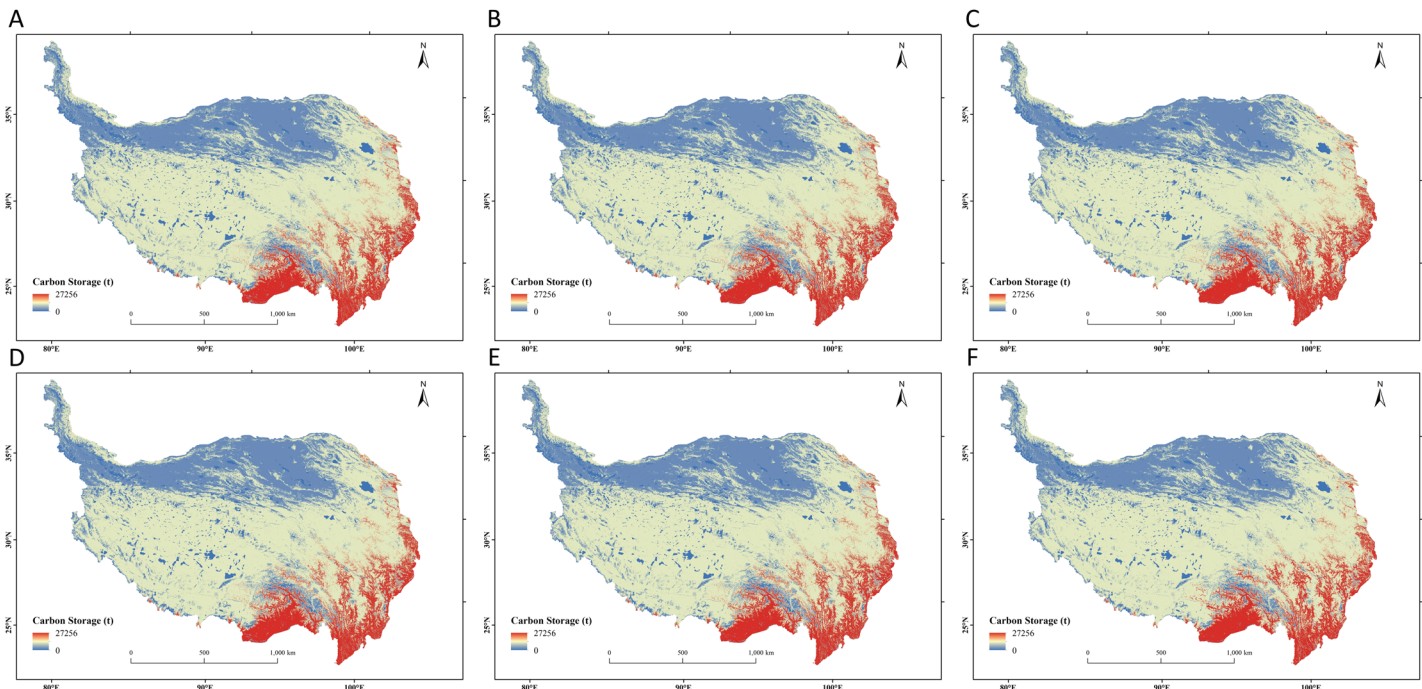

**Fig 9. Predicted distribution of carbon storage on the Qinghai-Tibet Plateau under different scenarios in 2030 and 2040.** (A) Natural Development Scenario in 2030; (B) Cropland Protection Scenario in 2030; (C) Ecological Protection Scenario in 2030; (D) Natural Development Scenario in 2040; (E) Cropland Protection Scenario in 2040; (F) Ecological Protection Scenario in 2040.

## 4 Discussion

### 4.1 Evaluation and analysis of carbon storage

As of 2020, the total carbon storage on the Qinghai-Tibet Plateau was approximately 25 Pg, with grasslands accounting for about 69.22%, which is generally consistent with the results of [36]. The conclusions of [42–44] show some discrepancies compared to ours (reaching 32 Pg, 43.56 Pg, and 42.36 Pg, respectively). These differences between studies may be due to variations in land use datasets and carbon density data. The spatial distribution of carbon storage shows a clear decreasing trend from southeast to northwest, with Nagqu in the central Qinghai-Tibet Plateau having the highest carbon storage. Between 2000 and 2020, carbon storage in the eastern Qinghai-Tibet Plateau increased significantly, while the central and western regions experienced declines. From 2020 to 2040, our study predicts an increase in carbon storage under three scenarios (natural development, farmland protection, and ecological conservation), with the fastest growth occurring under the ecological conservation scenario. This growth is primarily attributed to the expansion of forest and grassland areas, which play a key role in carbon sequestration, aligning closely with the findings of [45].

### 4.2 Discussion on the causes of spatial variation in carbon storage

The carbon storage in the Qinghai-Tibet Plateau exhibits a decreasing spatial distribution from southeast to northwest, closely related to the vegetation coverage, land use types, and

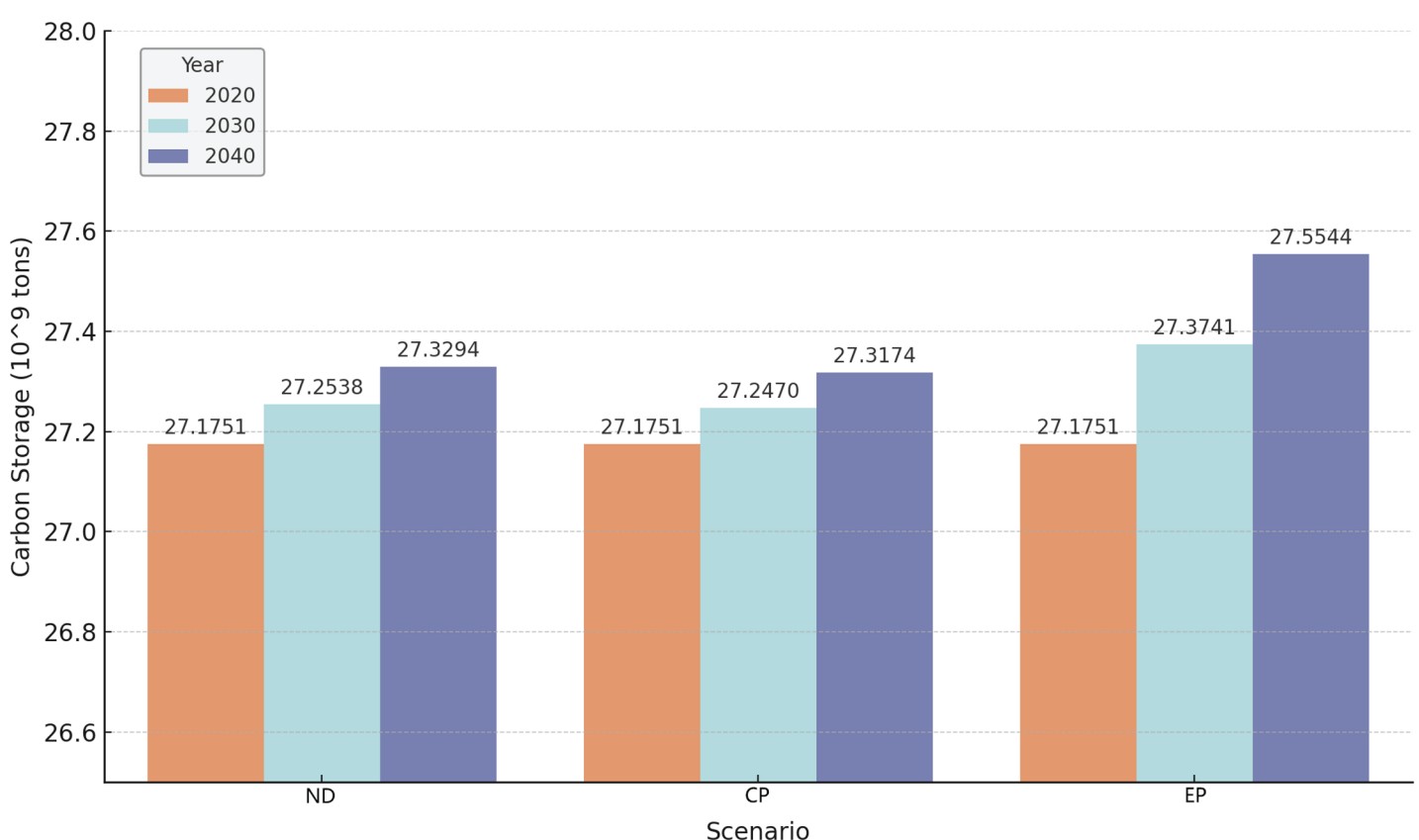

**Fig 10. Carbon storage changes on the Qinghai-Tibet Plateau under different scenarios from 2020 to 2040.**

climate conditions in the region. In the southeastern areas, such as Nyingchi and Garzê Tibetan Autonomous Prefecture, high forest coverage, diverse vegetation types, and relatively warm and humid climate create favorable conditions for carbon sequestration, particularly the high carbon density of forests significantly enhances the regional carbon storage. This phenomenon aligns with Shen et al.'s observation of forests as key carbon sinks [46].

In the central region of the Qinghai-Tibet Plateau (e.g., Nagqu), despite the cold climate, the extensive distribution of alpine meadows and higher vegetation coverage have still led to significant carbon accumulation, with soil organic carbon in grasslands serving as an important carbon storage source. This finding is consistent with the research by Xu et al. , which highlights the significant role of soil carbon in grasslands [47].

In contrast, the carbon storage in the northwestern region of the Qinghai-Tibet Plateau is notably lower. The climate in this area is dry and cold, with sparse vegetation primarily consisting of deserts, resulting in low carbon density and difficulties in effective carbon sequestration and storage. Wang et al. also observed that the extreme climate and barren land conditions in the northwest limit the enhancement of regional carbon storage [48]. The differences in soil quality and vegetation types further exacerbate the spatial imbalance in carbon storage distribution.

## 4.3 Discussion on the global factors influencing carbon storage

Based on the results of the OLS model, carbon storage on the Qinghai-Tibet Plateau is significantly influenced by multiple factors.

First, the DEM shows a significant positive impact on carbon storage, indicating that carbon sequestration capacity is stronger at higher altitudes on the plateau, possibly due to reduced human activity and a more intact ecosystem [49,50].

Annual average temperature also has a significant positive effect on carbon storage in the global model. While some studies have found that rising temperatures increase soil decomposition rates [51], research in extremely cold regions suggests that this positive correlation may result from the strong inhibition of soil organic carbon (SOC) decomposition at low temperatures, while net primary productivity (NPP) increases significantly with rising soil temperatures [52,53].

The aridity index shows a negative impact globally. Since the primary carbon storage type in the Tibetan Plateau is soil organic carbon, and grasslands and forests are significant carbon sources, under arid conditions, the vitality of plant roots decreases, leading to reduced organic matter input, which in turn inhibits the accumulation of soil organic carbon [54,55].

NDVI has a significant positive effect on carbon storage, reflecting the critical role of vegetation density in carbon sequestration. High NDVI values typically indicate dense and healthy vegetation, which enhances photosynthetic efficiency and increases carbon storage [32].

The distance from railways and roads also shows a positive effect on carbon storage, suggesting that areas farther from transportation infrastructure are more favorable for carbon accumulation. This may be because these areas are less disturbed by human activity, resulting in a more intact ecosystem [56].

Overall, topography, climate, vegetation density, and human activity collectively drive carbon storage capacity on the Qinghai-Tibet Plateau.

## 4.4 Discussion on local factors influencing carbon storage

According to the conclusions of Sect 3.2.3, carbon storage on the Qinghai-Tibet Plateau is influenced by the complex interaction of multiple factors.

First, the DEM has a particularly significant effect on carbon storage in the northwest of the plateau, indicating that carbon storage is highly sensitive to altitude. This finding is consistent with Hu et al.'s study on the northern slope of the Qilian Mountains, where they observed that soil organic carbon storage increases with elevation [57].

The influence of slope and aspect on carbon storage shows significant regional differences across the Qinghai-Tibet Plateau. In the southeastern region, steeper slopes are positively correlated with carbon storage, likely due to soil and water conservation measures, such as terracing and afforestation. However, in the western region, steep slopes generally have a negative impact on carbon storage, primarily due to natural conditions such as soil erosion and reduced vegetation. In these areas with complex terrain and significant elevation differences, sun-facing slopes typically have higher temperatures, which may promote carbon fixation. Yang et al. also noted that the inorganic carbon content on sun-facing slopes can be up to eight times that of shaded slopes [58].

The influence of the aridity index on carbon storage on the Qinghai-Tibet Plateau is also regionally significant. In the western regions of the Tibetan Plateau, the aridity index shows a significant positive correlation with carbon stocks. This is because the area has steep terrain, sparse vegetation cover, and relatively weak soil and water conservation capacity. If precipitation increases, the soil is prone to erosion, leading to lateral loss of soil organic carbon [59]. In contrast, in the relatively more humid and warmer eastern regions, an increase in the aridity index may reduce the effectiveness of plant photosynthesis, resulting in a decrease in carbon stocks. Related studies have found that vegetation in humid environments is more sensitive to drought events, and with the increasing frequency of droughts in the future, significant carbon loss may occur [60].

The economic structure of the Qinghai-Tibet Plateau is relatively simple, with residents' income primarily dependent on animal husbandry, forestry, and tourism. Therefore, in most areas, GDP is negatively correlated with carbon storage. However, in some central and southern areas, such as Shigatse, known as the "granary of Tibet," the favorable growing conditions and proper land management allow for economic crop cultivation that supports carbon storage accumulation. In regions where land resources and production methods are limited, population size and GDP are inversely related. In contrast, areas with low population density generally have relatively intact ecosystems and higher carbon storage.

In the northwest of the Qinghai-Tibet Plateau, dominated by alpine grasslands and deserts, vegetation is highly sensitive to water availability. Ecosystems that have adapted to long-term arid conditions may be adversely affected by a sudden increase in rainfall. On the other hand, in the southeastern forest and shrub regions, large trees and deep-rooted plants can effectively maintain soil structure and reduce carbon loss. Therefore, an increase in precipitation can, to some extent, promote the accumulation of soil organic carbon in these areas [58].

Temperature has a spatially variable effect on carbon storage on the Qinghai-Tibet Plateau. In the western, central, and northern regions, temperature is negatively correlated with carbon storage, likely because the vegetation in these regions is adapted to cold climates, and warming may exceed their tolerance, thereby limiting carbon fixation. At the same time, increased temperatures accelerate the decomposition of soil organic carbon, resulting in reduced carbon storage [61]. In contrast, temperature is positively correlated with carbon storage in the southeastern region, likely because the warmer climate is more suitable for plant growth, where temperature increases promote photosynthesis, enhancing carbon absorption and fixation, thereby offsetting some of the carbon loss from soil decomposition [62].

The proximity to railways and roads has a positive effect on carbon storage, and this effect is more pronounced in the southeastern part of the Qinghai-Tibet Plateau. Human activities associated with road construction and development along transportation corridors have led

to vegetation damage and land degradation, reducing carbon storage. In the northwest, where human activity is sparse and the ecosystem is relatively intact, the proximity to transportation infrastructure does not significantly negatively impact carbon storage, reflecting regional differences in the impact of transportation infrastructure on carbon storage.

On the Qinghai-Tibet Plateau, distance from water bodies also has varying effects on carbon storage in different regions. In the relatively arid northwestern and central regions, the farther the area is from water bodies, the lower the soil moisture content, and the weaker the ability of microbial decomposition to store carbon [63]. Moreover, the vegetation types in these regions have adapted to lower water requirements, and their growth and biochemical reactions are less influenced by water availability [64]. In contrast, in the warmer, wetter south-east, where vegetation is more dependent on water, sufficient water can promote plant growth and photosynthesis, thereby increasing the production and accumulation of organic matter [65]. Therefore, areas closer to water bodies have higher carbon stocks.

The relationship between NDVI and carbon storage varies significantly across different regions of the Qinghai-Tibet Plateau. In the northwest, NDVI is positively correlated with carbon storage. This region is primarily composed of alpine deserts and alpine grasslands, with sparse vegetation cover. Therefore, an increase in vegetation cover helps to enhance soil stability, thereby promoting carbon accumulation. In contrast, the main vegetation type in the eastern region is alpine meadow. However, climate warming may lead to a decrease in the stability of soil aggregates in these areas, driving the soil to release a large amount of $CO_2$. Even if the NDVI value increases, carbon stocks may still decrease [66,67].

## 4.5 Comprehensive analysis of global and local impacts

At the global level, OLS regression analysis reveals that average annual temperature, distance from railways, distance from roads, and NDVI have a general positive impact on carbon storage across the Qinghai-Tibet Plateau, while the aridity index serves as a negative explanatory variable. However, global analysis does not capture regional heterogeneity and cannot reveal the subtle variations in the influence of these drivers across different locations.

Using the GWR model, local analysis provides a better understanding of the spatial heterogeneity of driving factors. It shows that factors such as aspect, distance from water bodies, and average annual temperature have varying impacts on carbon storage across regions. For instance, the effect of aspect on carbon storage is especially significant in the northwestern part of the plateau, reflecting the strong control exerted by the complex terrain in this region. Additionally, the influence of distance from water bodies exhibits notable spatial variability, which is diminished and not captured in the global analysis. Moreover, local analysis reveals the significant impact of socioeconomic factors, such as GDP and population density, on carbon storage, which were not apparent in the global analysis. This indicates that local analysis better captures the specific mechanisms by which socioeconomic factors influence different areas. For example, in regions with higher GDP, more resources are allocated to environmental protection, which significantly enhances carbon storage. Conversely, in densely populated areas, changes in land use and vegetation degradation lead to a notable reduction in carbon storage.

In summary, global analysis provides an overall understanding of the drivers of carbon storage, highlighting the general influence of key environmental factors. In contrast, local analysis delves into the spatial variability of these drivers, particularly regarding the region-specific impacts of socioeconomic factors. Therefore, integrating the results of both

global and local analyses enables a more comprehensive understanding of the spatial distribution and driving mechanisms of carbon storage on the Qinghai-Tibet Plateau, thus providing stronger scientific support for developing regional and local carbon management strategies.

## 4.6 Future scenario predictions and management strategies

Based on predictions from the InVEST model, carbon storage on the Qinghai-Tibet Plateau shows an upward trend in the natural development, farmland protection, and ecological protection scenarios for 2030 and 2040. This trend is mainly influenced by several driving factors, including annual average temperature, proximity to railways, and aridity index. To further strengthen ecological protection, the following region-specific carbon management policy recommendations are proposed:

(a) **Optimization of land use management based on regional temperature-driven heterogeneity.** The impact of temperature variation on vegetation growth and carbon sequestration capacity exhibits significant spatial heterogeneity [68]. In the southeastern plateau (e.g., Nyingchi and Garzê), higher temperatures promote forest and grassland growth; thus, expanding vegetation cover and introducing high-biomass, adaptable species are recommended. Conversely, in high-altitude, colder regions (e.g., Nagqu and Chamdo), carbon storage relies on cold-resistant vegetation and stable soil carbon. Therefore, it is advised to introduce cold-tolerant plants, enhance soil management, and establish ecological buffer zones.

(b) **Strengthen ecological protection and restoration along railway and road corridors.** In ecologically fragile, high-traffic corridors (e.g., along the Lhasa-Nyingchi railway), establishing ecological buffer zones, implementing vegetation restoration, and creating ecological corridors are recommended to mitigate the impact of infrastructure on carbon storage. In remote, sparsely populated transportation corridors (e.g., along the Qinghai-Tibet Highway), localized ecological restoration measures should be implemented to enhance soil stability and increase carbon storage capacity.

(c) **Implement differentiated management strategies for areas with varying aridity indices.** In the arid northwestern regions (e.g., Ali and Nagqu areas), it is recommended that drought-resistant plants similar to local crops be introduced with the aim of enhancing the stability of carbon sinks. In contrast, in the relatively more humid eastern and northeastern regions, soil moisture conservation measures should be implemented, along with vegetation restoration projects. Furthermore, it is essential to monitor the long-term impacts of drought events on vegetation and carbon stocks, in order to facilitate prompt adjustments to strategies in response to intensifying droughts.

(d) **Promote ecological protection in low population density areas.** The population on the Qinghai-Tibet Plateau is mainly concentrated in Lhasa and Xining, while most areas have low population density [69]. In sparsely populated areas with limited human activity (e.g., Nagqu and Ali), establishing ecological protection zones and promoting eco-compensation policies and eco-tourism are recommended to balance carbon storage conservation and local economic development while preventing overdevelopment.

## 4.7 Uncertainty and limitations in implementation

Although this study reveals the significant relationship between carbon storage on the Qinghai-Tibet Plateau and land use types and climate factors, there are still some limitations.

First, the land use data and climate data used in this study cover a relatively short time span, failing to fully reflect long-term trends.

Second, when using the PLUS model to predict land use in the Qinghai-Tibet region, it relies on the Markov model to predict the demand for each land use type and then performs spatial allocation. However, this model primarily references historical data and does not fully consider factors such as socio-economic development, climate change, and policy changes. This leads to discrepancies between the simulation results and actual observations, which is a major limitation of this study. To improve prediction accuracy, key parameters such as socio-economic data, climate models, and policy impacts can be incorporated into the model [38]. The subjective influence on model parameter settings may introduce errors. Future research should focus on adaptive tuning of the model to enhance the reliability of simulation results. Since the PLUS model is still in the early stages of application, further optimization and refinement of this model have significant research potential, which could support land use planning and ecological protection decision-making.

Finally, the reliability of estimating carbon storage using the InVEST model largely depends on the accuracy of published Land Use/Land Cover Change (LUCC) products. Besides LUCC, the contribution of climate change to carbon storage dynamics is also significant. Since the InVEST model primarily relies on LUCC products as input, it can only reveal carbon storage dynamics related to LUCC, without fully accounting for the impact of climate change on carbon storage through other pathways. Climate change not only affects carbon storage by altering LUCC types but also influences the terrestrial carbon cycle by altering soil respiration and soil organic matter decomposition rates. Numerous studies have shown that the impact of climate change on the carbon cycle is multifaceted, but these factors have not been considered by the InVEST carbon model. Another debate regarding the use of the InVEST model for carbon storage assessment is that it only allows users to assign a uniform carbon density value for each land use type. This simplification may overlook the spatial heterogeneity within the same land use type. Additionally, the carbon storage changes caused by LUCC are a gradual process rather than an instantaneous change, which the InVEST model fails to reflect [24]. To more accurately assess carbon storage dynamics, future research should combine long-term observation data and dynamic models to better capture temporal changes in carbon storage. This will help improve the accuracy of model predictions and provide more reliable scientific support for carbon management and climate change adaptation strategies.

## 5 Conclusion

Based on the land use data of the Qinghai-Tibet Plateau from 2000 to 2020, the InVEST model was used to estimate the carbon storage on the Qinghai-Tibet Plateau, and the PLUS model was used to analyze and predict its spatiotemporal changes. The following conclusions were obtained:

(1) Between 2000 and 2020, the carbon storage on the Qinghai-Tibet Plateau exhibited a generally fluctuating upward trend, with the total carbon storage increasing by $1.4648 \times 10^8$ t, and soil organic carbon being the primary carbon pool.

(2) Grassland and forest land are the main sources of carbon storage, with grassland carbon storage averaging 69.39% of the total carbon storage and forest carbon storage accounting for 24.12%. The decrease in unused land and cropland areas led to a reduction in carbon storage, while the increase in water bodies and forest land elevated the overall carbon storage.

(3) Influencing factors: The global results from the OLS model indicate that NDVI is the primary factor influencing carbon storage, with a positive impact on carbon storage. Additionally, average annual temperature and human activities (such as proximity to

railways and roads) also significantly affect the distribution of carbon storage. The Geographically Weighted Regression (GWR) model further reveals the spatial heterogeneity of these driving factors, indicating that the influence of different factors on carbon storage varies significantly across regions.

(4) Spatial distribution characteristics: Carbon storage decreases from southeast to northwest, with higher carbon storage in the southeast where vegetation coverage is high and the climate is humid, and lower carbon storage in the northwest where the climate is dry and vegetation is sparse.

Through systematic analysis of carbon storage on the Qinghai-Tibet Plateau and its driving factors, this study provides scientific support for future land use planning and ecological protection policies, contributing to the sustainable development of the region.

## Author contributions

**Conceptualization:** Xu Chen.

**Data curation:** Lei Wang.

**Formal analysis:** Lei Wang.

**Funding acquisition:** Yaping Zhang, Xu Chen.

**Investigation:** Yaping Zhang.

**Validation:** Lei Wang.

**Visualization:** Lei Wang.

**Writing – original draft:** Lei Wang.

**Writing – review & editing:** Yaping Zhang, Xu Chen.

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
