## [Decision Letter · Decision Letter 0]

15 Oct 2024

PONE-D-24-41312Analysis and prediction of carbon storage changes on the Qinghai-Tibet PlateauPLOS ONE

Dear Dr. Zhang,

Thank you for submitting your manuscript to PLOS ONE. After careful consideration, we feel that it has merit but does not fully meet PLOS ONE’s publication criteria as it currently stands. Therefore, we invite you to submit a revised version of the manuscript that addresses the points raised during the review process.

We look forward to receiving your revised manuscript.

Kind regards,

Chun Liu

Academic Editor

PLOS ONE

Journal Requirements:

2. Please note that PLOS ONE has spec6ific guidelines on code sharing for submissions in which author-generated code underpins the findings in the manuscript. In these cases, all author-generated code must be made available without restrictions upon publication of the work. Please review our guidelines at https://journals.plos.org/plosone/s/materials-and-software-sharing#loc-sharing-code and ensure that your code is shared in a way that follows best practice and facilitates reproducibility and reuse.

“This research was funded by two sources. The first source of funding was the Yunnan Provincial Agricultural Basic Research Joint Special Project, supported by the Yunnan Provincial Science and Technology Department, with Grant No. 202101BD070001-042.

The second source of funding came from the Yunnan Ten-thousand Talents

Program, supported by the Yunnan Provincial Department of Human Resources and Social Security.”

5. We note that [S1,S2,S4,S6,S7 and S8 Figure] in your submission contain [map/satellite] images which may be copyrighted. All PLOS content is published under the Creative Commons Attribution License (CC BY 4.0), which means that the manuscript, images, and Supporting Information files will be freely available online, and any third party is permitted to access, download, copy, distribute, and use these materials in any way, even commercially, with proper attribution. For these reasons, we cannot publish previously copyrighted maps or satellite images created using proprietary data, such as Google software (Google Maps, Street View, and Earth). For more information, see our copyright guidelines: http://journals.plos.org/plosone/s/licenses-and-copyright.

a. You may seek permission from the original copyright holder of S1,S2,S4,S6,S7 and S8 Figure to publish the content specifically under the CC BY 4.0 license.  

6. Please upload a copy of Figure 1-8, to which you refer in your text on manuscript. If the figure is no longer to be included as part of the submission please remove all reference to it within the text.

7. We notice that your supplementary S1-S8 Figures are included in the manuscript file. Please remove them and upload them with the file type 'Supporting Information'. Please ensure that each Supporting Information file has a legend listed in the manuscript after the references list.

8. Please upload a copy of Supporting Information Figure/Table/etc. S1-S10 Tables which you refer to in your text on page 23.

Reviewers' comments:

Reviewer's Responses to Questions

**Comments to the Author**

1. Is the manuscript technically sound, and do the data support the conclusions?

Reviewer #1: Partly

Reviewer #2: Yes

2. Has the statistical analysis been performed appropriately and rigorously? 

Reviewer #1: I Don't Know

Reviewer #2: Yes

3. Have the authors made all data underlying the findings in their manuscript fully available?

Reviewer #1: Yes

Reviewer #2: Yes

4. Is the manuscript presented in an intelligible fashion and written in standard English?

Reviewer #1: Yes

Reviewer #2: Yes

5. Review Comments to the Author

Reviewer #1: The Qinghai-Tibet Plateau, as a crucial global carbon reservoir, plays a pivotal role in the carbon cycle. Studying changes in its carbon storage is of great significance for understanding global climate change. The paper uses the InVEST model and PLUS model to analyze changes in carbon storage on the Qinghai-Tibet Plateau and predicts future trends, demonstrating certain innovation and practicality in this research area. However, there is still room for improvement in research methods and results presentation, yet the paper provides certain reference value for studying carbon storage changes on the Qinghai-Tibet Plateau. There are still several issues that need to be addressed, particularly the inadequate discussion.

1. The quality of Figure 1 needs improvement.

2. Tables 2-4 could be merged into a single table or figure for better clarity.

3. The number of cited references throughout the paper is too low, representing a notable omission in this aspect.

4. The simulated data lacks comparison with other studies, and the parameters in Table 1 do not address vegetation and climate zoning.

5. The discussion section needs to be compared with previous research findings, which is the biggest issue of this paper. It tends to present the results in isolation without sufficient reference to existing literature. The discussion section is severely lacking in relevant references and needs to be reorganized.

Reviewer #2: 1 Gaps and research innovations to be filled by research need to be further clarified.

2 Explain why these variables and models were chosen for the uniqueness of the Tibetan Plateau.

3 Use more spatial distribution and trend plots to enhance the readability of the results section.

4 The heterogeneity of factors across regions needs to be clarified in terms of how carbon management policies can contribute to ecological conservation.

6. PLOS authors have the option to publish the peer review history of their article (what does this mean?). If published, this will include your full peer review and any attached files.

Reviewer #1: **Yes: **Lihui Luo

Reviewer #2: No

---

## [Author Response · Author response to Decision Letter 1]

2 Dec 2024

Dear Reviewers,

On behalf of all contributing authors, I would like to express our sincere appreciation for the reviewers' constructive comments on our article entitled “Analysis and Prediction of Carbon Storage Changes on the Qinghai-Tibet Plateau” (Manuscript No.: PONE-D-24-41312). We are grateful for the reviewers' valuable and helpful feedback on our manuscript. We have carefully reviewed the comments and made revisions accordingly. Based on the instructions provided in your letter, we have revised the manuscript, with modifications highlighted in yellow and blue—yellow indicating new content and blue indicating more standardized English expressions. Point-by-point responses to the and reviewers' comments are provided below this letter.

We would like to express our gratitude for the opportunity to resubmit a revised version of the manuscript and for your time and consideration.

Yours sincerely,

Yaping Zhang

Reviewer1:

Comments:

The Qinghai-Tibet Plateau, as a crucial global carbon reservoir, plays a pivotal role in the carbon cycle. Studying changes in its carbon storage is of great significance for understanding global climate change. The paper uses the InVEST model and PLUS model to analyze changes in carbon storage on the Qinghai-Tibet Plateau and predicts future trends, demonstrating certain innovation and practicality in this research area. However, there is still room for improvement in research methods and results presentation, yet the paper provides certain reference value for studying carbon storage changes on the Qinghai-Tibet Plateau. There are still several issues that need to be addressed, particularly the inadequate discussion.

Reply: The authors want to thank the reviewer for helpful comments. The authors have carefully revised these shortcomings in the manuscript.

1. The quality of Figure 1 needs improvement.

Reply 1: Thanks for the reviewer’s nice comment. The authors adjusted the colors in Figure 1 and revised the annotations for the regions, using abbreviations instead of full names to save unnecessary space and enhance the image's visual appeal and readability.

2. Tables 2-4 could be merged into a single table or figure for better clarity.

Reply 2: Thanks for the reviewer’s excellent comments. The authors have merged tables 2 to 4 into a new Table 2, improving clarity while reducing unnecessary space usage. The new table is placed in section 2.3 on page 5.

3. The number of cited references throughout the paper is too low, representing a notable omission in this aspect.

Reply 3: Thanks for the reviewer’s helpful comments. The authors have added relevant references in appropriate sections of the paper. These include works for comparison with our conclusions, papers supporting certain statements in the text, and references related to the selection and use of data.The newly added or modified reference numbers by the authors are 31-34, 38-39, and 41-68.

4.The simulated data lacks comparison with other studies, and the parameters in Table 1 do not address vegetation and climate zoning.

Reply 4: The authors thank the reviewer for their valuable comments. In response to the reviewer’s suggestions, we have conducted a detailed evaluation and analysis of the simulated data, comparing it with data from related studies, and have revised the discussion section accordingly. We have added Section 4.1 (on page 14) to assess the carbon storage simulation data. Additionally, in Sections 4.2 to 4.4 (formerly Sections 4.1 to 4.3), we further analyze the impact of various driving factors on carbon storage by comparing our results with those of other studies.

5. The discussion section needs to be compared with previous research findings, which is the biggest issue of this paper. It tends to present the results in isolation without sufficient reference to existing literature. The discussion section is severely lacking in relevant references and needs to be reorganized.

Reply 5: The authors sincerely thank the reviewer for the valuable comments. In response, we have thoroughly revised and reorganized the discussion section to better compare and contextualize our findings with previous research. Relevant conclusions have been analyzed in relation to existing literature, and modifications have been made throughout Sections 4.1 to 4.7. The specific modifications have been highlighted in the text.

Reviewer2:

1. Gaps and research innovations to be filled by research need to be further clarified.

Reply 1: The authors appreciate the reviewer’s valuable comments. Following the reviewer’s suggestions, we have reorganized the introduction to provide a more detailed discussion on the importance of studying carbon storage changes and their driving factors, as well as the limitations of existing research. Additionally, the authors have summarized the contributions and research objectives of this study, emphasizing how it addresses the shortcomings of current methods. We revised the wording in the second paragraph of the introduction on page 2, merged and modified the third and fourth paragraphs, and summarized the contributions and research objectives in the final paragraph of the introduction.

2. Explain why these variables and models were chosen for the uniqueness of the Tibetan Plateau.

Reply 2:The authors want to thank the reviewer for the nice comments. In terms of variable selection, the ecosystem of the Qinghai-Tibet Plateau is highly fragile, impacted by climate change (e.g., rising temperatures, changes in precipitation, and increasing aridity) and human activities (e.g., overgrazing and industrial development). This vulnerability makes ecological restoration especially crucial. Therefore, selecting drivers related to ecosystem health—such as soil properties and vegetation cover—can help researchers identify key factors influencing restoration, facilitating the development of more effective restoration strategies. Furthermore, the ecosystem services of the Qinghai-Tibet Plateau—such as water conservation, soil retention, and eco-tourism—are essential for regional sustainable development. By incorporating variables linked to ecosystem services, such as proximity to railways or roads, researchers can comprehensively evaluate the enhancement of ecological service value when assessing the cost-effectiveness of restoration projects, providing scientific support for restoration efforts. The plateau’s complex topography, with significant slope variability, further exacerbates the impact of slope aspects. Therefore, including topographic factors like slope and aspect in the analysis is essential.

Regarding model selection, the high altitude and low temperatures of the Qinghai-Tibet Plateau make large-scale ground surveys costly and labor-intensive. On the other hand, remote sensing lacks the precision of ground-level details, posing challenges to accurately assessing carbon storage in the region. Model simulations offer a cost-effective solution, providing broader spatial and temporal coverage, integrating multi-source data, and supporting complex decision-making analyses.

Compared to other models, the PLUS (Patch-generating Land Use Simulation) model is particularly suited for handling complex land-use changes. It combines multiple random seeds with cellular automata (CA), offering higher simulation accuracy. This is especially important for ecologically diverse regions like the Qinghai-Tibet Plateau, which face multiple pressures. The model can reveal the driving mechanisms behind land-use changes, aiding in the formulation of more precise restoration strategies. Additionally, given the Qinghai-Tibet Plateau's role as a significant global carbon sink, the selection of the InVEST model for carbon storage assessment highlights its considerable ecological value and the critical role of carbon storage in climate regulation and ecosystem stability, enhancing our understanding of ecological potential and challenges under future scenarios. OLS fitting can provide an overall trend analysis, but given the complexity of the Qinghai-Tibet Plateau's geography, as well as its significant climate and ecosystem diversity, OLS is insufficient to capture local variations. Different regions' ecosystems respond uniquely to climate, topography, and human activities, resulting in diverse impacts across areas. GWR, on the other hand, assigns different regression coefficients to each observation point spatially, allowing the analysis to reflect localized characteristics for different regions. This approach is well-suited to the Qinghai-Tibet Plateau, where geographic and climate conditions are complex and vary significantly across regions. GWR can reveal which driving factors, such as altitude, temperature, or NDVI, have a more substantial impact on carbon storage or ecosystem services in specific areas, providing a more refined basis for management.

The authors have added explanations in Section 2.2 (Dataset) on page 4 to clarify the reasons for selecting relevant variables. Additionally, further details on model selection have been included in Sections 2.3.2 to 2.3.4.

The authors have added explanations in Section 2.2 (Dataset) on page 4 to clarify the reasons for selecting the relevant variables.

The description of the PLUS model in Section 2.3.2 on page 5 has been revised to clarify the specific reasons for selecting this model.

Similarly, the description of the InVEST model in Section 2.3.3 on page 6 has been revised to explain the rationale for selecting this model, with the specific modifications highlighted in the text.

In Section 2.3.4 on page 8, the authors have added explanations for selecting the OLS and GWR models to conduct relevant research on the Qinghai-Tibet Plateau.

Due to the limited research on land use prediction for the Tibetan Plateau and the variability introduced by different land use datasets, there may be some discrepancies when comparing simulation data with other studies. The land use data in this study is based on the Chinese annual land cover dataset (CLCD) created by Professor Huang Xin at Wuhan University using Landsat data, with an overall accuracy of 80%. Through model accuracy validation, it can be demonstrated that the future land use data simulated by the PLUS model is reliable. The authors added content regarding model validation in Section 2.3.2 on page 6.

3. Use more spatial distribution and trend plots to enhance the readability of the results section.

Reply 3: The authors want to thank the reviewer for the useful comments. The authors added a land use change figure (Fig. 8), illustrating the transition relationships between various land use types from 2020 to 2040, in Section 3.3.1 on page 12. The specific modifications have been highlighted in the text.

The authors added a total carbon storage change figure (Fig. 10), illustrating the changes in total carbon storage on the Qinghai-Tibet Plateau from 2020 to 2040, in Section 3.3.2 on page 14. The specific modifications have been highlighted in the text.

4. The heterogeneity of factors across regions needs to be clarified in terms of how carbon management policies can contribute to ecological conservation.

Reply 4: The authors want to thank the reviewer for helpful comments. The authors reorganized the section 4.7 (page 21) on future scenario predictions and management strategies to better reflect the impact of different regional driving factors on carbon storage in the Qinghai-Tibet Plateau, thereby providing targeted and regionally heterogeneous policy recommendations. The specific modifications have been highlighted in the text.

---

## [Decision Letter · Decision Letter 1]

21 Jan 2025

PONE-D-24-41312R1Analysis and prediction of carbon storage changes on the Qinghai-Tibet PlateauPLOS ONE

Dear Dr. Zhang,

Thank you for submitting your manuscript to PLOS ONE. After careful consideration, we feel that it has merit but does not fully meet PLOS ONE’s publication criteria as it currently stands. Therefore, we invite you to submit a revised version of the manuscript that addresses the points raised during the review process.

We look forward to receiving your revised manuscript.

Kind regards,

Chun Liu

Academic Editor

PLOS ONE

Journal Requirements:

Reviewers' comments:

Reviewer's Responses to Questions

**Comments to the Author**

1. If the authors have adequately addressed your comments raised in a previous round of review and you feel that this manuscript is now acceptable for publication, you may indicate that here to bypass the “Comments to the Author” section, enter your conflict of interest statement in the “Confidential to Editor” section, and submit your "Accept" recommendation.

Reviewer #1: All comments have been addressed

Reviewer #2: All comments have been addressed

2. Is the manuscript technically sound, and do the data support the conclusions?

Reviewer #1: Yes

Reviewer #2: Yes

3. Has the statistical analysis been performed appropriately and rigorously? 

Reviewer #1: Yes

Reviewer #2: Yes

4. Have the authors made all data underlying the findings in their manuscript fully available?

Reviewer #1: Yes

Reviewer #2: Yes

5. Is the manuscript presented in an intelligible fashion and written in standard English?

Reviewer #1: Yes

Reviewer #2: Yes

6. Review Comments to the Author

Reviewer #1: The authors have adequately addressed my concerns in the previous round of review, I have no further commetns and recommend the publication of this article in PLOS ONE.

Reviewer #2: 1 The article has been significantly improved in content, structure and methodology, especially in the Results and Discussion section, where many valuable analyses have been added.

2 Focus on core findings and reduce redundancy.

7. PLOS authors have the option to publish the peer review history of their article (what does this mean?). If published, this will include your full peer review and any attached files.

Reviewer #1: No

Reviewer #2: No

---

## [Author Response · Author response to Decision Letter 2]

29 Jan 2025

Dear reviewer:

On behalf of all the authors, I would like to express my sincere gratitude to the reviewers for their affirmation of our article titled "Analysis and Prediction of Carbon Stock Changes in the Qinghai Tibet Plateau" (manuscript number: PONE-D-24-41312R1). We greatly appreciate the valuable and beneficial feedback from the reviewers on our manuscript. According to the instructions in your letter, we have revised the manuscript and highlighted the modified parts in yellow and blue - yellow represents new content and blue represents more standard English expression. Below this letter is a point by point response to the reviewer's comments.

We greatly appreciate the opportunity to resubmit the revised manuscript and thank you for your time and consideration.

Yours sincerely,

Yaping Zhang

Editor：

Reply: The authors want to thank the editor for their important suggestions. The authors checked the references and made modifications to some of them. [1] is a special report with an incorrect citation format, and the authors have replaced it with an article.

1.Kerr, Richard A. Global warming is changing the world. American Association for the Advancement of Science. 2007;316(5822):188–190.

[6] is an online article, the authors have modified the format of the references.

6. Nixon J. The economic impact of global warming. An Oxford Economics White Paper. 2020. Available from:

https://www.cheshirescouts.org.uk/wp-content/uploads/The-economic-impact of-global-warming-Oxford-Economics.pdf

The author information abbreviation in reference [11] is incorrect, and the authors have corrected it.

11. Chen GS, Tian HQ. Land use/cover change effects on carbon cycling in terrestrial ecosystems. Chinese Journal of Plant Ecology. 2007;31(2):189.

The abbreviation of the publishing house information in reference [18] is incorrect, and the authors have made changes to it.

18. MacDicken KG, et al. A guide to monitoring carbon storage in forestry and agroforestry projects. Winrock International Institute for Agricultural Development, USA; 1997.

[33], [37], [55], [58], and [67] are Chinese papers, and the authors used (in Chinese) for annotation at the end and made modifications to the presentation of author information.

33. Wang JN, Wang WC, Hai MM. Simulation and Analysis of Land Use Change in Shandong Province Based on the PLUS Model. Territory and Natural Resources Research. 2022;(06):1–8. (in Chinese)

37. He Q. Ecosystem Service Function Assessment and Multi-Scenario Prediction of the Eastern Hubei Region Based on the InVEST Model. Hubei Normal University; 2023. (in Chinese)

55. Xu C, Ruan HH, Wu XQ, Xie YC, Yang Y. Research Progress on the Impact of

 Drought on Forest Soil Organic Carbon Turnover and Accumulation. Journal of

 Nanjing Forestry University (Natural Sciences Edition). 2022;46(6):195. (in

 Chinese)

58. Yang F, Zhang GL, Huang LM, Li DC, Yang F, et al. Vertical Distribution

Characteristics and Influencing Factors of Soil Organic Carbon and Inorganic

Carbon in the Topographic Sequence of Alpine Mountain Regions. Acta Pedologica Sinica. 2015;52(06):1226–1236. (in Chinese)

67. Li N, Wang GX, Gao YH, Ji CZ. Research Progress on Soil Organic Carbon in the Ecosystems of the Qinghai-Tibet Plateau. Soils. 2009;41(4):512–519. (in Chinese)

The authors have made modifications to the author information representation methods in [35] and [48].

35. Zhang YL. Integration dataset of Tibet Plateau boundary; 2019. Available from:

https://dx.doi.org/10.11888/Geogra.tpdc.270099.

48. Wang L, Zeng H, Zhang YJ, Zhao G, Chen N, Li JX. A review of research on soil

 carbon storage and its influencing factors in the Tibetan Plateau. Chinese Journal of Ecology. 2019;38(11):3506.

Reviewer1:

Comments:

The authors have adequately addressed my concerns in the previous round of review, I have no further comments and recommend the publication of this article in PLOS ONE.

Reply: The author wants to express their gratitude to the reviewer for their affirmation.

Reviewer2:

1.The article has been significantly improved in content, structure and methodology, especially in the Results and Discussion section, where many valuable analyses have been added.

2. Focus on core findings and reduce redundancy.

Reply: The authors would like to express their gratitude to the reviewer for their comment.

---

## [Editor Report · Decision Letter 2]

13 Feb 2025

Analysis and prediction of carbon storage changes on the Qinghai-Tibet Plateau

PONE-D-24-41312R2

Dear Dr. Zhang,

We’re pleased to inform you that your manuscript has been judged scientifically suitable for publication and will be formally accepted for publication once it meets all outstanding technical requirements.

Kind regards,

Chun Liu

Academic Editor

PLOS ONE
---

## [Editor Report · Acceptance letter]

PONE-D-24-41312R2

PLOS ONE

Dear Dr. Zhang,

I'm pleased to inform you that your manuscript has been deemed suitable for publication in PLOS ONE. Congratulations! Your manuscript is now being handed over to our production team.

Kind regards,

on behalf of

Dr. Chun Liu

Academic Editor

PLOS ONE